# Algorithm and Hardness for Dynamic Attention Maintenance in Large Language Models

## Abstract

Large language models (LLMs) have made fundamental changes in human life. The attention scheme is one of the key components over all the LLMs, such as BERT, GPT-1, Transformers, GPT-2, 3, 3.5 and 4. Inspired by previous theoretical study of static version of the attention multiplication problem [Zandieh, Han, Daliri, and Karbasi ICML 2023, Alman and Song NeurIPS 2023]. In this work, we formally define a dynamic version of attention matrix multiplication problem. There are matrices $Q, K, V \in \mathbb{R}^{n \times d}$, they represent query, key and value in LLMs. In each iteration we update one entry in $K$ or $V$. In the query stage, we receive $(i, j) \in [n] \times [d]$ as input, and want to answer $(D^{-1}AV)_{i,j}$, where $A := \exp(QK^\top) \in \mathbb{R}^{n \times n}$ is a square matrix and $D := \mathrm{diag}(A\mathbf{1}_n) \in \mathbb{R}^{n \times n}$ is a diagonal matrix. Here $\mathbf{1}_n$ denote a length-$n$ vector that all the entries are ones.

We provide two results: an algorithm and a conditional lower bound.

- On one hand, inspired by the lazy update idea from [Demetrescu and Italiano FOCS 2000, Sankowski FOCS 2004, Cohen, Lee and Song STOC 2019, Brand SODA 2020], we provide a data-structure that uses $O(n^{\omega(1,1,\tau)-\tau})$ amortized update time, and $O(n^{1+\tau})$ worst-case query time, where $n^{\omega(1,1,\tau)}$ denotes $\mathcal{T}_{\mathrm{mat}}(n, n, n^\tau)$ with matrix multiplication exponent $\omega$ and $\tau$ denotes a constant in $(0, 1]$.
- On the other hand, show that unless the hinted matrix vector multiplication conjecture [Brand, Nanongkai and Saranurak FOCS 2019] is false, there is no algorithm that can use both $O(n^{\omega(1,1,\tau)-\tau-\Omega(1)})$ amortized update time, and $O(n^{1+\tau-\Omega(1)})$ worst query time.

In conclusion, our algorithmic result is conditionally optimal unless hinted matrix vector multiplication conjecture is false.

One notable difference between prior work [Alman and Song NeurIPS 2023] and our work is, their techniques are from the area of fine-grained complexity, and our techniques are not. Our algorithmic techniques are from recent work in convex optimization, e.g. solving linear programming. Our hardness techniques are from the area of dynamic algorithms.

## 1 Introduction

Large language models (LLMs) such as Transformer Vaswani et al. (2017), BERT Devlin et al. (2018), GPT-3 Brown et al. (2020), PaLM Chowdhery et al. (2022), and OPT Zhang et al. (2022a) offer better results when processing natural language compared to smaller models or traditional techniques. These models possess the capability to understand and produce complex language, which is beneficial for a wide range of applications like language translation, sentiment analysis, and question answering. LLMs can be adjusted to multiple purposes without requiring them to be built from scratch. A prime example of this is ChatGPT, a chat software developed by OpenAI utilizing GPT-3's potential to its fullest. GPT-4 OpenAI (2023), the latest iteration, has the potential to surpass the already impressive abilities of GPT-3, including tasks such as language translation, question answering, and text generation. As such, the impact of GPT-4 on NLP could be significant, with new applications potentially arising in areas like virtual assistants, chatbots, and automated content creation.

The primary technical foundation behind LLMs is the attention matrix Vaswani et al. (2017); Radford et al. (2018); Devlin et al. (2018); Brown et al. (2020). Essentially, an attention matrix is a square matrix with corresponding rows and columns representing individual words or "tokens," and entries indicating their correlations within a given text. This matrix is then utilized to gauge the essentiality of each token in a sequence, relative to the desired output. As part of the attention mechanism, each input token is assigned a score or weight based on its significance or relevance to the current output, which is determined by comparing the current output state and input states through a similarity function.

More formally, the attention matrix can be expressed as follows: Suppose we have two matrices, $Q$ and $K$, comprising query and key tokens respectively, where $Q \in \mathbb{R}^{n \times d}$ and $K \in \mathbb{R}^{n \times d}$. The attention matrix is a square $n \times n$ matrix denoted by $A$ that relates the input tokens in the sequence. After normalizing using the softmax function, each entry in this matrix quantifies the attention weight or score between a specific input token (query token $Q$) and an output token (key token $K$). Notably, entries along the diagonal reflect self-attention scores, indicating the significance of each token in relation to itself.

When modeling long sequences with large $n$, the most significant hindrance to accelerating LLM operations is the duration required for carrying out attention matrix calculations Kitaev et al. (2020); Wang et al. (2020). These calculations involve multiplying the attention matrix $A$ with another value token matrix $V \in \mathbb{R}^{n \times d}$. In Wang et al. (2020), they demonstrate that the self-attention mechanism can be approximated by a low-rank matrix. They propose a new self-attention mechanism and used it in their Linformer model. In Kitaev et al. (2020), they replace dot-product attention with one that uses locality-sensitive hashing, which also improves the time complexity.

Furthermore, the static attention computation and approximation has been studied by Alman & Song (2023) from both algorithmic and hardness perspectives. However, in practice, the attention matrix needs to be trained and keeps changing. In this work, we study the dynamic version of the attention computation problem. By using a dynamic approach, the attention weights can be updated on-the-fly as new information is introduced, enabling the model to adapt more effectively to changes in the input. This is particularly beneficial in cases where the input data is highly dynamic and subject to frequent changes, such as in natural language processing applications where the meaning and context of words and phrases can be influenced by the surrounding text.

Following the prior work Zandieh et al. (2023); Alman & Song (2023), we formally define the standard attention computation problem as follows. To distinguish their standard model with the dynamic version studied in this paper, we call the problem defined in Zandieh et al. (2023); Alman & Song (2023) "static" version of attention multiplication. Another major difference between previous work Zandieh et al. (2023); Alman & Song (2023) and our work is that they studied an approximate version, whereas we study the exact version.

**Definition 1.1** (Static Attention Multiplication). *Given three matrices $Q, K, V \in \mathbb{R}^{n \times d}$, we define attention computation $\mathsf{Att}(Q, K, V) = D^{-1}AV$ where square matrix $A \in \mathbb{R}^{n \times n}$ and diagonal matrix $D \in \mathbb{R}^{n \times n}$ are $A := \exp(QK^\top), D := \mathrm{diag}(A\mathbf{1}_n)$. Here we apply the $\exp(\cdot)$ function entry-wise[1]. We use $\mathbf{1}_n$ to denote a length-$n$ vector where all the entries are ones. The $\mathrm{diag}()$ function is taking a length-$n$ vector as input and outputs an $n \times n$ diagonal matrix by copying that vector on the diagonal of the output matrix. See Figure 1 and Figure 2 for an illustration.*

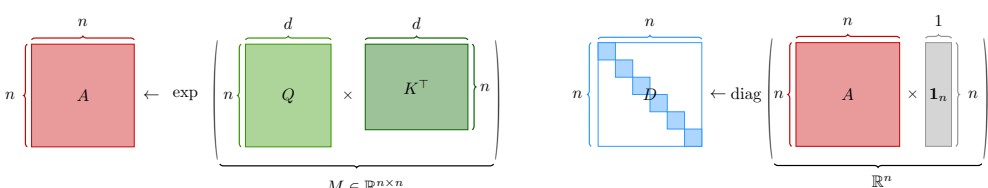

Figure 1: Computation of the attention matrix $A = \exp(QK^\top)$ and the diagonal matrix $D \in \mathbb{R}^{n \times n}$ (defined in Definition 1.1). Here $\exp()$ is the entry-wise function.

---

[1] For a matrix $M \in \mathbb{R}^{n \times n}$, following the transformer literature, we use $\exp(M)_{i,j} := \exp(M_{i,j})$.

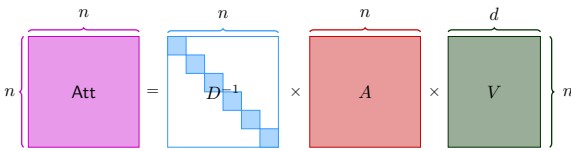

Figure 2: Computation of the target matrix $\text{Att}(Q, K, V) = D^{-1}AV$ (defined in Definition 1.1)

In applied LLMs training, the model parameters are changing slowly during training Chen et al. (2021). In addition, deep neural network architectures frequently exhibit significant redundancy, and empirical evidence supports the capacity of deep neural networks to tolerate substantial levels of sparsity Han et al. (2015); Gale et al. (2019). In downstream fine-tuning tasks, the dimensions of the model often make the fine-tuning infeasible. Over the past few years, numerous techniques for inducing sparsity have been proposed to sparsify the neural network such as magnitude pruning Zhu & Gupta (2017), RegL Evci et al. (2020) and dynamic sparse reparameterization Mostafa & Wang (2019). Thus, it is worth considering the dynamic version of Attention multiplication problem which update the attention matrix entry-wise. Next, we formally define the "dynamic" or "online" version of attention multiplication problem, we call it ODAMV[2]. For consistency of the discussion, we will use the word "online" in the rest of the paper.

**Definition 1.2** (ODAMV$(n, d)$). *The goal of **O**nline **D**iagonal-based normalized **A**ttention **M**atrix **V**ector multiplication problem* ODAMV$(n, d)$ *is to design a data-structure that satisfies the following operations:*

1. INIT*: Initialize on three $n \times d$ matrices $Q$, $K$, $V$.*

2. UPDATE*: Change any entry of $K$, or $V$.*

3. QUERY*: For any given $i \in [n]$, $j \in [d]$, return $(D^{-1} \exp(QK^\top)V)_{i,j}$.*

   - *Here $D := \text{diag}(\exp(QK^\top)\mathbf{1}_n) \in \mathbb{R}^{n \times n}$ is a positive diagonal matrix.*
   - *Here $[n]$ denotes the set $\{1, 2, \cdots, n\}$.*

In this paper, we first propose a data-structure that efficiently solves the ODAMV problem (Definition 1.2) by using lazy update techniques. We then complement our result by a conditional lower bound. On the positive side, we use lazy update technique in the area of dynamic algorithms to provide an upper bound. In the area of theoretical computer science, it is very common to assume some conjecture in complexity when proving a lower bound. For example, P $\neq$ NP, (strong) exponential time hypothesis, orthogonal vector and so on Abboud & Williams (2014); Henzinger et al. (2015); Backurs & Indyk (2015); Backurs et al. (2017); Chen (2018); Rubinstein (2018); Alman et al. (2020; 2023); Alman & Song (2023). To prove our conditional lower bound, we use a conjecture which is called **H**inted **M**atrix **V**ector multiplication (HMV) conjecture (Brand et al., 2019, Conjecture 5.2). On the negative side, we show a lower bound of computing solving ODAMV assuming the HMV conjecture holds.

### 1.1 OUR RESULTS

We first show our upper bound result making use of the lazy update strategy.

**Theorem 1.3** (Upper bound, informal version of Theorem B.1). *For any constant $a \in (0, 1]$. Let $d = O(n)$. Let $\delta \in \mathbb{R}$ denote the update to the matrix. There is a dynamic data structure that uses $O(n^2)$ space and supports the following operations:*

- INIT$(Q, K, V)$. *It runs in $O(\mathcal{T}_{\text{mat}}(n, n, n))$ time.*[3]

---

[2]The name of our problem is inspired by a well-known problem in theoretical computer science which is called **O**nline **M**atrix **V**ector multiplication problem (OMV) Henzinger et al. (2015); Larsen & Williams (2017); Chakraborty et al. (2018).

[3]We use $\mathcal{T}_{\text{mat}}(n, d, m)$ to denote the time of multiplying a $n \times d$ matrix with another $d \times m$ matrix. For more details, we refer the readers to Section 2.

- UPDATEK$(i \in [n], j \in [d], \delta \in \mathbb{R})$. *This operation updates one entry in $K$, and it runs in $O(\mathcal{T}_{\mathrm{mat}}(n, n^a, n)/n^a)$ amortized[4] time.*

- UPDATEV$(i \in [n], j \in [d], \delta \in \mathbb{R})$. *This operation takes same amortized[4] time as UPDATEK.*

- QUERY$(i \in [n], j \in [d])$. *This operation outputs $(D^{-1}(\exp(QK^\top))V)_{i,j}$ and takes $O(n^a)$ worst-case time.*

The parameter $a$ allows for a trade-off between update and query time. For example, $a = 1$ leads to $O(n^{1.373})$ update time and $O(n)$ query time whereas $a = 1/2$ leads to $O(n^{1.55})$ update and $O(\sqrt{n})$ query time, using current bounds on $\mathcal{T}_{\mathrm{mat}}(\cdot, \cdot, \cdot)$ Alman & Williams (2021); Gall & Urrutia (2018). We remark that our results beat the naive $O(n^2)$ update time regardless of which fast matrix multiplication algorithm is used[5]. E.g., when using Strassen's algorithm Strassen et al. (1969) we get an update time of $O(n^{2+(1.8075-2)a})$.

Our second result makes use of a variation of the popular online matrix vector multiplication (OMV) conjecture which is called hinted matrix vector multiplication conjecture (see Definition C.2 and Brand et al. (2019)). Next, we present a lower bound for the problem of dynamically maintaining the attention computation Att$(Q, K, V)$ that matches our upper bound from Theorem 1.3.

**Lemma 1.4** (Lower bound, informal version of Lemma C.5). *Assuming the HMV conjecture is true. For every constant $0 < \tau \leq 1$, there is no algorithm that solves the ODAMV$(n, d)$ problem (see formal version in Definition C.4) with*

- *polynomial initialization time, and*

- *amortized update time $O(\mathcal{T}_{\mathrm{mat}}(n, n^\tau, d)/n^{\tau+\Omega(1)})$, and*

- *worst query time $O(n^{\tau-\Omega(1)})$.*

Conditional lower bounds identify the nature/origin of the hardness. E.g., problems with hardness from the OV (orthogonal vector) conjecture Williams (2005); Abboud et al. (2014) boil down to the fundamental bottleneck of searching, hardness from the BMM (boolean matrix multiplication) conjecture Abboud & Williams (2014) show that hardness comes from matrix multiplication, and problems with hardness from the HMV conjecture boil down to the trade-off between matrix-vector multiplication vs fast matrix multiplication. We show that dynamic attention maintenance belongs to the latter class by providing tight upper and conditional lower bounds.

## 1.2 RELATED WORK

**Static Attention Computation** A recent work by Zandieh, Han, Daliri, and Karbasi Zandieh et al. (2023) was the first to give an algorithm with provable guarantees for approximating the attention computation. Their algorithm makes use of locality sensitive hashing (LSH) techniques Charikar et al. (2020). They show that the computation of partition functions in the denominator of softmax function can be reduced to a variant of the kernel density estimation (KDE) problem, and an efficient KDE solver can be employed through subsampling-based swift matrix products. They propose the KDEformer which can approximate the attention within sub-quadratic time and substantiated with provable spectral norm bounds. In contrast, earlier findings only procure entry-wise error bounds. Based on empirical evidence, it was confirmed that KDEformer outperforms other attention approximations in different pre-trained models, in accuracy, memory, and runtime.

In another recent work Alman & Song (2023), they focus on the long-sequence setting with $d = O(\log n)$. The authors established that the existence of a fast algorithm for approximating the attention computation is dependent on the value of $B$, given the guarantees of $\|Q\|_\infty \leq B$, $\|K\|_\infty \leq B$, and $\|V\|_\infty \leq B$. They derived their lower bound proof by building upon a different line of work that dealt with the fine-grained complexity of KDE problems, which was previously studied in

---

[4]We remark that the presented data structure can be made worst-case via standard techniques (sometimes referred to as "global rebuilding") from the dynamic algorithm area Overmars (1983); Sankowski (2004); Goranci et al. (2017); Frandsen & Frandsen (2009).

[5]This is because $\mathcal{T}_{\mathrm{mat}}(n, n^a, n) \leq n^{2+(\omega-2)a}$.

Backurs et al. (2017); Alman et al. (2020). Their proof was based on a fine-grained reduction from the Approximate Nearest Neighbor search problem ANN. Additionally, their findings explained how LLM computations can be made faster by assuming that matrix entries are bounded or can be well-approximated by a small number of bits, as previously discussed in Zafrir et al. (2019), Section 2 and Katharopoulos et al. (2020), Section 3.2.1. Specifically, they Alman & Song (2023) showed a lower bound stating that when $B \geq \Omega(\sqrt{\log n})$, there is no algorithm that can approximate the computation in subquadratic time. However, when $B < o(\sqrt{\log n})$, they proposed an algorithm that can approximate the attention computation almost linearly.

**Transformer Theory**   Although the achievements of transformers in various fields are undeniable, there is still a significant gap in our precise comprehension of their learning mechanisms. Although these models have been examined on benchmarks incorporating numerous structured and reasoning activities, comprehending the mathematical aspects of transformers still considerably lags behind. Prior studies have posited that the success of transformer-based models, such as BERT Devlin et al. (2018), can be attributed to the information contained within its components, specifically the attention heads. These components have been found to hold a significant amount of information that can aid in solving various probing tasks related to syntax and semantics, as noted by empirical evidence found in several studies Hewitt & Manning (2019); Clark et al. (2019); Tenney et al. (2019); Hewitt & Liang (2019); Vig & Belinkov (2019); Belinkov (2022).

Various recent studies have delved into the representational power of transformers and have attempted to provide substantial evidence to justify their expressive capabilities. These studies have employed both theoretical as well as controlled experimental methodologies through the lens of Turing completeness Bhattamishra et al. (2020b), function approximation Yun et al. (2020), formal language representation Bhattamishra et al. (2020a); Ebrahimi et al. (2020); Yao et al. (2021), abstract algebraic operation learning Zhang et al. (2022b), and statistical sample complexity Wei et al. (2021); Edelman et al. (2022) aspects. According to the research conducted by Yun et al. (2020), transformers possess the capability of functioning as universal approximators for sequence-to-sequence operations. Similarly, the studies carried out by Pérez et al. (2019); Bhattamishra et al. (2020b) have demonstrated that attention models may effectively imitate Turing machines. In addition to these recent works, there have been several previous studies that aimed to assess the capacity of neural network models by testing their learning abilities on simplistic data models Siegelmann & Sontag (1992); Yao et al. (2021); Zhang et al. (2022b). Furthermore, Li et al. (2023a) conducted a formal analysis of the training dynamics to further understand the type of knowledge that the model learns from such data models. According to findings from a recent study Zhao et al. (2023), moderately sized masked language models have demonstrated the ability to parse with satisfactory results. Additionally, the study utilized BERT-like models that were pre-trained using the masked language modeling loss function on the synthetic text generated with probabilistic context-free grammar. They empirically validated that these models can recognize syntactic information that aids in partially reconstructing a parse tree. Li et al. (2023b) studied the computation of regularized version of exponential regression problem (without normalization factor). In Zhang et al. (2023); Liu et al. (2023), they speedup the inference time from both theoretical perspective and experimental perspective by leverage the property of attention. In Wu et al. (2023), they develop an information-theoretic framework that formulates soft prompt tuning as maximizing mutual information between prompts and other model parameters.

**Dynamic Maintenance**   In recent years, projection maintenance has emerged as a crucial data structure problem. The effectiveness and efficiency of several cutting-edge convex programming algorithms greatly hinge upon a sturdy and streamlined projection maintenance data structure Cohen et al. (2019); Lee et al. (2019); Brand (2020); Jiang et al. (2020b); Brand et al. (2020); Jiang et al. (2021); Song & Yu (2021); Brand (2021); Jiang et al. (2020a); Huang et al. (2022); Gu & Song (2022). There are two major differences between the problem in the dynamic data structure for optimization and our dynamic attention matrix maintenance problem. The first notable difference is that, in the optimization task, the inverse of a full rank square matrix is typically computed, whereas, in the attention problem, we care about the inverse of a positive diagonal matrix which behaves the normalization role in LLMs. The second major difference is, in the standard optimization task, all the matrix matrix operations are linear operations. However, in LLMs, non-linearity such as softmax/exp function is required to make the model achieve good performance. Therefore, we need to apply an entry-wise nonlinear function to the corresponding matrix. In particular, to compute $f(QK^\top)V$

when $f$ is linear function, we can pre-compute $K^\top V$. However when $f$ is $\exp$ function, we are not allowed to compute $K^\top V$ directly.

Next, we will give more detailed reviews for classical optimization dynamic matrix maintenance problems. Let $B \in \mathbb{R}^{m \times n}$, consider the projection matrix $P = B^\top (BB^\top)^{-1} B$. The projection maintenance problem asks the following data structure problem: it can preprocess and compute an initial projection. At each iteration, $B$ receives a low rank or sparse change, and the data structure needs to update $B$ to reflect these changes. It will then be asked to approximately compute the matrix-vector product, between the updated $P$ and an online vector $h$. For example, in linear programming, one sets $B = \sqrt{W} A$, where $A \in \mathbb{R}^{m \times n}$ is the constraint matrix and $W$ is a diagonal matrix. In each iteration, $W$ receives relatively small perturbations. Then, the data structure needs to output an approximate vector to $\sqrt{W} A^\top (AWA^\top)^{-1} A \sqrt{W} h$, for an online vector $h \in \mathbb{R}^n$.

**Roadmap**  The rest of the paper is organized as follows. In Section 2, we give some preliminaries. In Section 3, we explain the techniques used to show our upper bound and lower bound results. In Section 4, we provide a lower bound proof for the simplified version of dynamic attention problem. In Section 5, we provide the conclusion for our paper. We defer the full proofs of upper bound in Appendix B. We defer the full proofs of lower bound in Appendix C.

## 2  PRELIMINARY

For a matrix $A$, we use $A^\top$ to denote its transpose. For a matrix $A$, use $A_{i,j}$ to denote its entry at $i$-th row and $j$-th column. For a non-zero diagonal matrix $D \in \mathbb{R}^{n \times n}$, we use $D^{-1} \in \mathbb{R}^{n \times n}$ to denote the matrix where the $(i, i)$-th diagonal entry is $(D_{i,i})^{-1}$ for all $i \in [n]$. For a vector $x \in \mathbb{R}^n$, we use $\mathrm{diag}(x) \in \mathbb{R}^{n \times n}$ to denote an $n \times n$ matrix where the $i, i$-th entry on the diagonal is $x_i$ and zero everywhere else for all $i \in [n]$. We use $\exp(M)$ to denote the entry-wise exponential, i.e., $\exp(M)_{i,j} := \exp(M_{i,j})$. We use $\mathbf{1}_n$ to denote the length-$n$ vector where all the entries are ones. We use $\mathbf{0}_n$ to denote the length-$n$ vector where all entries are zeros.

We define a standard notation for describing the running time of matrix multiplication.

**Definition 2.1.** *For any three positive integers, we use $\mathcal{T}_{\mathrm{mat}}(a, b, c)$ to denote the time of multiplying an $a \times b$ matrix with another $b \times c$ matrix.*

We use $\omega$ to denote the time that $n^\omega = \mathcal{T}_{\mathrm{mat}}(n, n, n)$. Currently $\omega \approx 2.373$ Williams (2012); Le Gall (2014); Alman & Williams (2021).

**Definition 2.2.** *We define $\omega(\cdot, \cdot, \cdot)$ function as follows, for any $a, b$ and $c$, we use $\omega(a, b, c)$ to denote that $n^{\omega(a,b,c)} = \mathcal{T}_{\mathrm{mat}}(n^a, n^b, n^c)$.*

## 3  TECHNIQUE OVERVIEW

Given three matrices $Q, K, V \in \mathbb{R}^{n \times d}$, we need to compute the attention given by $\mathrm{Att}(Q, K, V) = D^{-1} A V$ where square matrix $A \in \mathbb{R}^{n \times n}$ and diagonal matrix $D \in \mathbb{R}^{n \times n}$ are $A := \exp(QK^\top)$, $D := \mathrm{diag}(A\mathbf{1}_n)$. The static problem Alman & Song (2023) is just computing $\mathrm{Att}$ for given $Q, K$ and $V$. In the dynamic problem, we can get updates for $K$ and $V$ in each iteration.

Due to space limitation, we only describe the core ideas and proof sketch of upper bound in Section 3.1. For the complete proofs, we refer the readers to read the Appendix B. Similarly, we only give high description for lower bound in Section 3.2 and defer the details into Appendix C.

### 3.1  ALGORITHM

**Problem Formulation**  For each update, we receive $\delta$ as input and update one entry in either matrix $K$ or $V$. In the query function, we take index $i \in [n], j \in [d]$ as input, and return the $\{i, j\}$-th element in the target matrix $B := D^{-1} A V$.

Let $C$ denote $AV$. Let $\widetilde{B}$ denote the updated target matrix $B$. We notice that the computation of the attention can be written as $\widetilde{B} = (D^{-1} + \Delta_D)(C + \Delta_C)$. Let $\Delta^{(t)}$ denote the change in the $t$-th

iteration. In a lazy-update fashion, we write $\widetilde{B}$ in the implicit form

$$\widetilde{B} = (D^{-1} + \sum_{t=1}^{ct} \Delta_D^{(t)})(C + \sum_{t=1}^{ct} \Delta_C^{(t)})$$

where ct denotes the number of updates since the last time we recomputed $D$ and $C$.

**Lazy Update**   We propose a lazy-update algorithm (Algorithm 2) that does not compute the attention matrix when there is an update on the key matrix $K$. We also propose a lazy-update algorithm (Algorithm 3) that does not compute the attention matrix when there is an update on the value matrix $V$. Instead, we maintain a data-structure (Algorithm 1) that uses $\text{List}_C, \text{List}_D$ and $\text{List}_V$ to record the update by storing rank-1 matrices before the iteration count reaches the threshold $n^a$ for some constant $a$. For the initialization (Algorithm 1), we compute the exact target matrix $D^{-1}AV$ and other intermediate matrices, which takes $O(\mathcal{T}_{\text{mat}}(n, d, n))$ time (Lemma B.3).

**Re-compute**   When the iteration count reaches the threshold $n^a$, we re-compute all the variables in the data-structure as follows (Lemma B.8). By using Fact A.1, we first stack all the rank-1 matrices in $\text{List}_C$ and compute the matrix multiplication once to get $\sum_{t=1}^{ct} \Delta_C^{(t)}$ using $\mathcal{T}_{\text{mat}}(n, n^a, d) = n^{\omega(1,1,a)}$ time (Lemma B.9). Then, we compute $C + \sum_{t=1}^{ct} \Delta_C^{(t)}$ to get the re-computed $\widetilde{C}$. Similarly, to re-compute $V$, we stack all the rank-1 matrices in $\text{List}_V$ and compute the matrix multiplication once to get $\sum_{t=1}^{ct} \Delta_V^{(t)}$ using $\mathcal{T}_{\text{mat}}(n, n^a, d) = n^{\omega(1,1,a)}$ time. Then, we compute $V + \sum_{t=1}^{ct} \Delta_V^{(t)}$ to get the re-computed $\widetilde{V}$. To re-compute the diagonal matrix $D$, we sum up all the updates by $\sum_{t=1}^{ct} \Delta_D^{(t)}$ and add it to the old $D^{-1}$ (detail can be found in Algorithm 5). Hence, our algorithm takes $n^{\omega(1,1,a)}/n^a$ amortized time to update $K$ and $V$ (Lemma B.4, Lemma B.5).

**Fast Query**   Recall that the query function takes index $i \in [n], j \in [d]$ as input, and returns the $\{i, j\}$-th element in the target matrix $B := D^{-1}AV$. Let $\widetilde{D}^{-1}$ denote the lates $D^{-1}$ obtained from $\text{List}_D$. Let $\Delta_{V,1}$ and $\Delta_{V,2}$ be stacked matrix obtained from list from $V$. We can rewrite the output by

$$((\widetilde{D}^{-1}) \cdot (A) \cdot (V + \Delta_{V,1}\Delta_{V,2}))_{i,j} = ((\widetilde{D}^{-1}) \cdot (A \cdot V))_{i,j} + ((\widetilde{D}^{-1}) \cdot A \cdot (\Delta_{V,1}\Delta_{V,2}))_{i,j}$$
$$= (\widetilde{D})_i^{-1}(C_{i,j} + (\Delta_{C,1}\Delta_{C,2})_{i,j}) + (\widetilde{D})_i^{-1}A_{i,*}\Delta_{V,1}(\Delta_{V,2})_{*,j}.$$

Note that we maintain $C$ in our re-compute function. Hence, computing the first part takes $O(n^a)$ time. As each column of $\Delta_{V,1}$ and row of $\Delta_{V,2}$ is 1-sparse, computing the second part takes $O(n^a)$ time. The total running time needed for the query function is $O(n^a)$ (Lemma B.7, Lemma B.6).

## 3.2   HARDNESS

We now turn to our lower bound result, which is inspired by the HMV conjecture (Brand et al., 2019, Conjecture 5.2). Let us firstly define the HMV problem (see formal definition in Definition C.2).

Let the computation be performed over the boolean semi-ring. For any $0 < \tau \le 1$, the HMV problem has the following three phases

- **Phase 1.** Input two $n \times n$ matrices $M$ and $V$
- **Phase 2.** Input an $n \times n$ matrix $P$ with at most $n^\tau$ non-zero entries
- **Phase 3.** Input a single index $i \in [n]$
    - We need to answer $MPV_{*,i}$
    - Here $V_{*,i} \in \mathbb{R}^n$ is the $i$-th column of matrix $V$

According to Brand et al. (2019), the above problem is conjectured to be hard in the following sense,

**Conjecture 3.1** (Hinted MV (HMV), (Brand et al., 2019, Conjecture 5.2))**.** *For every constant* $0 < \tau \le 1$ *no algorithm for the hinted Mv problem (Definition C.2) can simultaneously satisfy*

- *polynomial time in* **Phase 1.**

- $O(n^{\omega(1,1,\tau)-\epsilon})$ *time complexity in* **Phase 2.** *and*

- $O(n^{1+\tau-\epsilon})$ in **Phase 3.**

*for some constant $\epsilon > 0$.*

Our primary contribution lies in demonstrating how to reduce HMV problem (Definition C.2) to OAMV (Definition 4.1) and ODAMV (Definition C.4). To achieve this, we have adopted a contradiction-based approach. Essentially, we begin by assuming the existence of an algorithm that can solve the OAMV problem with polynomial initialization time and amortized update time of $O(\mathcal{T}_{\mathrm{mat}}(n, n^\tau, d)/n^{\tau+\Omega(1)})$, while worst-case query time is $O(n^{\tau-\Omega(1)})$ for all $\tau \in (0, 1]$. Our assumption implies that there exists a data structure that is faster than our result (Theorem B.1). We subsequently proceed to demonstrate that using this algorithm enables us to solve the HMV problem too quickly, which contradicts the HMV conjecture.

Specifically, let us take an instance for the HMV problem (Definition C.2)

- Let $\mathsf{M}, \mathsf{V} \in \{0, 1\}^{n \times n}$ denote two matrices from **Phase 1.** from HMV.

We create a new instance $\mathsf{OAMV}(\widetilde{n} = n, \widetilde{d} = n)$ where $\widetilde{Q} = \mathsf{M}, \quad \widetilde{K} = 0, \quad \widetilde{V} = \mathsf{V}$.

In Claim 4.3 and Claim 4.4, by making use of our construction of $\widetilde{Q}, \widetilde{K}$ and $\widetilde{V}$, we show that for each $i \in [n]$ and $j \in [n]$,

$$\text{If } ((\exp(\widetilde{Q}\widetilde{K}^\top) - \mathbf{1}_{n \times n})\widetilde{V})_{j,i} > 0, \text{ then } (\mathsf{MPV})_{j,i} = 1.$$
$$\text{If } ((\exp(\widetilde{Q}\widetilde{K}^\top) - \mathbf{1}_{n \times n})\widetilde{V})_{j,i} = 0, \text{ then } (\mathsf{MPV})_{j,i} = 0.$$

By using the above two statements, we know that $\exp(\widetilde{Q}\widetilde{K}^\top)\widetilde{V}_{*,i}$ is enough to reconstruct $\mathsf{MPV}_{*,i}$ for the HMV problem (Definition C.2). Then, solving $\mathsf{MPV}_{*,i}$ takes polynomial initialization time and amortized update time of $O(\mathcal{T}_{\mathrm{mat}}(n, n^\tau, d)/n^{\tau+\Omega(1)})$, while worst-case query time is $O(n^{\tau-\Omega(1)})$ for every $\tau \in (0, 1]$. The contradiction of the HMV conjecture shows that there is no such algorithm. Similarly, for the normalized case ODAMV (Definition C.4) problem, we show how to reconstruct another instance of the HMV problem and complete the proof by contradiction.

## 4 THE LOWER BOUND FOR A SIMPLIFIED VERSION

We define the dynamic attention matrix vector problem here. For the following definition, we ignore the effect by the normalization factor for simplicity. We will show how to handle the normalization factor in the Appendix (see Appendix C).

**Definition 4.1** ($\mathsf{OAMV}(n, d)$). *The goal of the **O**nline **A**ttention **M**atrix **V**ector Multiplication problem $\mathsf{OAMV}(n, d)$ is to design a data structure that satisfies the following operations:*

1. INIT: *Initialize on $n \times d$ matrices $Q$, $K$, $V$.*

2. UPDATE: *Change any entry of $Q$, $K$, or $V$.*

3. QUERY: *For any given $i \in [n]$, $j \in [d]$, return $(\exp(QK^\top)V)_{i,j}$.*

Next, we present our lower bound result ignoring the normalization factor.

**Lemma 4.2.** *Assuming the hinted $Mv$ conjecture (Conjecture C.3): For every constant $0 < \tau \leq 1$, there is no dynamic algorithm for $\mathsf{OAMV}(n, d)$ problem (Definition 4.1) with*

- *polynomial initialization time, and*

- *amortized update time $O(\mathcal{T}_{\mathrm{mat}}(n, n^\tau, d)/n^{\tau+\Omega(1)})$, and*

- *worst query time $O(n^{\tau-\Omega(1)})$.*

*Proof.* Assume there was a dynamic algorithm faster than what is stated in Lemma 4.2 for some parameter $\tau$, i.e. update time $O(\mathcal{T}_{\mathrm{mat}}(n, n^\tau, d)/n^{\tau+\epsilon})$ and query time $O(n^{\tau-\epsilon})$ for some constant $\epsilon > 0$. We show that this would contradict the hinted $Mv$ conjecture (Conjecture C.3).

Let us take an instance for the $v$-hinted Mv problem (Definition C.2) with $\mathsf{M}, \mathsf{V} \in \{0, 1\}^{n \times n}$. We create a new instance $\mathsf{OAMV}(\widetilde{n} = n, \widetilde{d} = n)$ where

$$\widetilde{Q} = \mathsf{M}, \quad \widetilde{K} = 0, \quad \widetilde{V} = \mathsf{V}$$

During phase 1, we give this input to the dynamic algorithm for the OAMV problem (Definition 4.1). During phase 2, when we receive the $n \times n$ matrix $\mathsf{P}$ with $n^\tau$ non-zero entries, we perform $n^\tau$ updates to the data structure to set $\widetilde{K}^\top = \mathsf{P}$. This time is bounded by

$$O(\widetilde{n}^\tau \cdot (\mathcal{T}_{\mathrm{mat}}(\widetilde{n}, \widetilde{n}^\tau, \widetilde{d})/\widetilde{n}^{\tau+\epsilon})) = O(n^{\omega(1,1,\tau)-\epsilon}).$$

At last, in phase 3, we perform $\widetilde{n}$ queries to obtain the column $\exp(\widetilde{Q}\widetilde{K}^\top)\widetilde{V}_{*,i}$ in $O(\widetilde{n} \cdot \widetilde{n}^{\tau-\epsilon}) = O(n^{1+\tau-\epsilon})$ time.

Using Claim 4.3, and Claim 4.4, we know that $\exp(\widetilde{Q}\widetilde{K}^\top)\widetilde{V}_{*,i}$ is enough to reconstruct $\mathsf{MPV}_{*,i}$ for the hinted $Mv$ problem. □

**Claim 4.3.** *For each $i \in [n]$ and $j \in [n]$, if $((\exp(\widetilde{Q}\widetilde{K}^\top) - \mathbf{1}_{n \times n})\widetilde{V})_{j,i}$ is $> 0$, then $(\mathsf{MPV})_{j,i} = 1$,*

*Proof.* Assume we have $((\exp(\widetilde{Q}\widetilde{K}^\top) - \mathbf{1}_{n \times n})\widetilde{V})_{j,i} > 0$, We defined $\widetilde{Q} = \mathsf{M}, \widetilde{K} = \mathsf{P}, \widetilde{V} = \mathsf{V}$, so we can rewrite it as $((\exp(\mathsf{MP}) - \mathbf{1}_{n \times n})\mathsf{V})_{j,i} > 0$. Using the definition of matrix multiplication, and the fact that $\exp(x) > 1$ for all $x > 0$, we have some $k \in [n]$ with

$$((\exp(\mathsf{MP}) - \mathbf{1}_{n \times n})_{j,k}(\mathsf{V})_{k,i} > 0$$
$$((\exp(\mathsf{MP})_{j,k} - 1)(\mathsf{V})_{k,i} > 0$$

We can conclude that for each $i \in [n], j \in [n]$, there is at least one $k \in [n]$ such that $\mathsf{V}_{k,i} > 0$ and $(\mathsf{MP})_{j,k} > 0$. Therefore, by using the definition of boolean semi-ring, we can conclude that $(\mathsf{MPV})_{j,i} = 1$ □

**Claim 4.4.** *For each $i \in [n]$ and $j \in [n]$, if $((\exp(\widetilde{Q}\widetilde{K}^\top) - \mathbf{1}_{n \times n})\widetilde{V})_{j,i}$ is $0$ then $(\mathsf{MPV})_{j,i} = 0$.*

*Proof.* We have

$$((\exp(\widetilde{Q}\widetilde{K}^\top) - \mathbf{1}_{n \times n})\widetilde{V})_{j,k} = ((\exp(\widetilde{Q}\widetilde{K}^\top) - \mathbf{1}_{n \times n}))_{j,*}\widetilde{V}_{*,i} = ((\exp(\mathsf{MP}) - \mathbf{1}_{n \times n}))_{j,*}\mathsf{V}_{*,i}$$

where the first step follows from the definition of matrix multiplication and the second step follows from the definition of $\widetilde{Q}, \widetilde{K}$ and $\widetilde{V}$.

By using the above equation, if $((\exp(\widetilde{Q}\widetilde{K}^\top) - \mathbf{1}_{n \times n})\widetilde{V})_{j,k} = 0$, we have

$$(\exp(\mathsf{MP}) - \mathbf{1}_{n \times n})_{j,*}\mathsf{V}_{*,i} = 0 \tag{1}$$

Eq. (1) implies that, for all $k \in [n]$ such that $\mathsf{V}_{k,i} = 1$, we have $(\exp(\mathsf{MP}) - \mathbf{1}_{n \times n})_{j,k} = 0$, which also implies that $(\mathsf{MP})_{j,k} = 0$.

Now, we can conclude that $(\mathsf{MPV})_{j,i} = 0$ for each $i \in [n]$ and $j \in [n]$. □

## 5 CONCLUSION

The development of Large Language Models (LLMs) has had a profound impact on society, with the attention mechanism being a critical aspect of LLMs. This study introduces the dynamic version of the attention matrix multiplication and delivers two outcomes - an algorithm and a conditional lower bound. The algorithmic outcome presents a data structure that supports the dynamic maintenance of attention computations, with a $O(n^{\omega(1,1,\tau)-\tau})$ amortized update time, and $O(n^{1+\tau})$ worst-case query time. The lower bound illustrates that the algorithm is conditionally optimal unless the conjecture on hinted matrix vector multiplication is incorrect. It is an interesting future direction to prove an unconditional lower bound. The problem of dynamic attention matrix multiplication, as proposed, focuses on updating only one entry at a time in either the $K$ or $V$ matrix during each iteration. It is possible to update multiple entries simultaneously in both matrices in practice. Therefore, further research could expand the scope of the problem formulation to include such situations. To the best of our knowledge, our research is purely theoretical and does not appear to have any negative societal impact that should be noted.

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
