APPENDIX

**Roadmap.**

In Section A, we provide several basic notations, definitions and more related work. In Section B, we present our dynamic data-structure. Our algorithm shows the upper bound results. In Section C, we give our conditional lower bound result by assuming the Hinted MV conjecture.

## A PRELIMINARY

**Notations** For a matrix $A$, we use $A^\top$ to denote its transpose. For a non-zero diagonal matrix $D \in \mathbb{R}^{n \times n}$, we use $D^{-1} \in \mathbb{R}^{n \times n}$ to denote the matrix where the $(i, i)$-th diagonal entry is $(D_{i,i})^{-1}$ for all $i \in [n]$.

For a vector $x \in \mathbb{R}^n$, we use $\mathrm{diag}(x) \in \mathbb{R}^{n \times n}$ to denote an $n \times n$ matrix where the $i, i$-th entry on the diagonal is $x_i$ and zero everywhere else for all $i \in [n]$.

In many theoretical computer science (TCS)/machine learning (ML) literature, $\exp(M)$ denotes the matrix exponential, i.e., $\exp(M) = \sum_{i=0}^\infty \frac{1}{i!} M^i$. However, in this paper, we use $\exp(M)$ to denote the entry-wise exponential, i.e.,

$$\exp(M)_{i,j} := \exp(M_{i,j}).$$

We use $\mathbf{1}_n$ to denote the length-$n$ vector where all the entries are ones. We use $\mathbf{0}_n$ to denote the length-$n$ vector where all entries are zeros.

In this work, we use standard notation $\mathcal{T}_{\mathrm{mat}}(\cdot, \cdot, \cdot)$ (see Definition 2.1) and $\omega(\cdot, \cdot, \cdot)$ (see Definition 2.2) for describing the running time of matrix multiplication, see literature Demetrescu & Italiano (2000); Zwick (2002); Sankowski (2004; 2005); Le Gall (2014); Brand & Nanongkai (2019); Cohen et al. (2019); Lee et al. (2019); Brand et al. (2019); Brand (2020); Gu & Ren (2021); Jiang et al. (2021); Brand (2021) for examples.

We give a standard fact that is used in our proof.

**Fact A.1** (folklore). *Given a set of vectors $a_1, \cdots, a_k \in \mathbb{R}^n$ and $b_1, \cdots b_k \in \mathbb{R}^d$, then we have $\sum_{i=1}^k a_i b_i^\top = AB^\top$ where $A \in \mathbb{R}^{n \times k}$ and $a_i$ is $i$-th column of $A$, and $B \in \mathbb{R}^{d \times k}$ and $b_i$ is the $i$-th column of $B$ for all $i \in [k]$. Further, we have*

- *Part 1. Computing $AB^\top$*

  - *takes $O(nkd)$ time, if we do it naively*
  - *takes $\mathcal{T}_{\mathrm{mat}}(n, k, d)$ time, if we use fast matrix multiplication*

- *Part 2. For any matrix $C \in \mathbb{R}^{d \times d}$, computing $AB^\top C$*

  - *takes $\mathcal{T}_{\mathrm{mat}}(n, k, d) + \mathcal{T}_{\mathrm{mat}}(n, d, d)$, if we use fast matrix multiplication, first compute $AB^\top$ then compute $(AB^\top)C$*
  - *takes $\mathcal{T}_{\mathrm{mat}}(k, d, d) + \mathcal{T}_{\mathrm{mat}}(n, k, d)$ time, if we use fast matrix multiplication, first compute $B^\top C$, then compute $A(B^\top C)$*

**Detailed Comparison with Alman & Song (2023)** In Alman & Song (2023), from the upper bound side, they make use of the 'polynomial method in algorithm design'. The polynomial method is a technique for finding low-rank approximations of $f(M)$, where M is a matrix and f is an entry-wise function. They apply a polynomial method to decompose $\exp(QK^\top)$ to $U_1 U_2$, where $U_1$ and $U_2$ are low rank matrices. Hence, for the follow-up attention computation (i.e., $\exp(QK^\top)V$) , they can first compute $U_2 V$, and then compute $U_1(U_2 V)$. As $U_1$ and $U_2$ are low rank matrices, these two steps can be computed efficiently. From the lower bound perspective, they give a fine-grained reduction from the Approximate Nearest Neighbor search (ANN) to attention problems. The hypothesis uses the Strong exponential time hypothesis.

In our case, from the upper bound side, we first proposed a data-structure that efficiently solves the Online Diagonal-based normalized Attention Matrix Vector multiplication problem by using the lazy

update techniques. Instead of updating the target matrix every time, we set a hyperparameter $a$ that lets the user strike the balance between the query time and the update time. From the lower bound side, we make use of a variation of the popular online matrix vector multiplication conjecture which is called hinted matrix vector multiplication conjecture. Notably, our work achieves congruence between upper and lower bound results for dynamically maintaining attention computations.

## B  MAIN UPPER BOUND

In Section B.1, we show the running time of initializing our data structure. In Section B.2, we show the running time of updating $K$ and $V$. In Section B.3, we show the correctness and the running time of querying the target matrix. In Section B.4, we show the correctness and the running time of recomputing the variables in our data-structure.

We propose our upper bound result as the following:

**Theorem B.1** (Main algorithm, formal version of Theorem 1.3)**.** *For any constant $a \in (0, 1]$. Let $d = O(n)$. There is a dynamic data structure that uses $O(n^2)$ space and supports the following operations:*

- INIT$(Q, K, V)$. *It runs in $O(\mathcal{T}_{\mathrm{mat}}(n, d, n))$ time.*

- UPDATEK$(i \in [n], j \in [d], \delta \in \mathbb{R})$. *This operation updates one entry in $K$, and it runs in $O(\mathcal{T}_{\mathrm{mat}}(n, n^a, n)/n^a)$ amortized time.*

- UPDATEV$(i \in [n], j \in [d], \delta \in \mathbb{R})$. *This operation takes same amortized time as $K$ update.*

- QUERY$(i \in [n], j \in [d])$. *This operation outputs $(D^{-1}(\exp(QK^\top))V)_{i,j}$ operation takes in $O(n^a)$ worst case time.*

**Remark B.2.** *The amortized time in* UPDATEK *and* UPDATEV *can be made into worst case time by using standard techniques, e.g. see Section B of Brand et al. (2019).*

---

**Algorithm 1** Dynamic Data Structure

---
 1: **data structure** DYNAMICATTENTION $\qquad\qquad\qquad\qquad\qquad\qquad$ ▷ Theorem B.1
 2: **members**
 3: $\quad$ $Q \in \mathbb{R}^{n \times d}$ $\qquad\qquad\qquad\qquad\qquad\qquad\qquad\qquad\qquad\qquad\qquad$ ▷ Query token
 4: $\quad$ $K \in \mathbb{R}^{n \times d}$ $\qquad\qquad\qquad\qquad\qquad\qquad\qquad\qquad\qquad\qquad\qquad\quad$ ▷ Key token
 5: $\quad$ $V \in \mathbb{R}^{n \times d}$ $\qquad\qquad\qquad\qquad\qquad\qquad\qquad\qquad\qquad\qquad\qquad$ ▷ Value token
 6: $\quad$ $M \in \mathbb{R}^{n \times n}$ $\qquad\qquad\qquad\qquad\qquad\qquad\qquad$ ▷ The logits matrix, $M = QK^\top$
 7: $\quad$ $A \in \mathbb{R}^{n \times n}$ $\qquad\qquad\qquad\qquad\qquad\qquad$ ▷ The attention matrix, $A = \exp(QK^\top)$
 8: $\quad$ $D \in \mathbb{R}^{n \times n}$ $\qquad\qquad\qquad\qquad\qquad\qquad\qquad\qquad\qquad$ ▷ The diagonal matrix,
 9: $\quad$ $C \in \mathbb{R}^{n \times d}$ $\qquad\qquad\qquad\qquad\qquad$ ▷ Intermediate matrix, $C = \exp(QK^\top)V$
10: $\quad$ $B \in \mathbb{R}^{n \times d}$ $\qquad\qquad\qquad\qquad\qquad\qquad$ ▷ Target matrix, $B = D^{-1}AV$
11: $\quad$ List$_A$ $\qquad\qquad\qquad\qquad\qquad\qquad\qquad\qquad\qquad\qquad\qquad$ ▷ List with size $n^a$
12: $\quad$ List$_C$ $\qquad\qquad\qquad\qquad\qquad\qquad\qquad\qquad\qquad\qquad\qquad$ ▷ List with size $n^a$
13: $\quad$ List$_D$ $\qquad\qquad\qquad\qquad\qquad\qquad\qquad\qquad\qquad\qquad\qquad$ ▷ List with size $n^a$
14: $\quad$ $\mathrm{ct}_K, \mathrm{ct}_V$
15: **end members**
16:
17: **procedure** INIT$(Q, K, V)$ $\qquad\qquad\qquad\qquad\qquad\qquad\qquad\qquad\qquad$ ▷ Lemma B.3
18: $\quad$ $Q \leftarrow Q, K \leftarrow K, V \leftarrow V$
19: $\quad$ $M \leftarrow QK^\top, A \leftarrow \exp(QK^\top)$
20: $\quad$ $C \leftarrow \exp(QK^\top)V$
21: $\quad$ $B \leftarrow D^{-1}AV$
22: $\quad$ $\mathrm{ct}_K \leftarrow 0$
23: $\quad$ $\mathrm{ct}_V \leftarrow 0$
24: **end procedure**
25: **end data structure**

---

---

**Algorithm 2** Algorithm that update $K$ and maintain the data structure

---

1: **data structure** DYNAMICATTENTION          ▷ Theorem B.1
2: **procedure** UPDATEK($i \in [n], j \in [d], \delta$)          ▷ Lemma B.4
3:      $\mathrm{ct}_K \leftarrow \mathrm{ct}_K + 1$
4:      $\widetilde{K}_{i,j} \leftarrow K_{i,j} + \delta$
5:      $(\Delta_M)_{*,i} \leftarrow \delta \cdot \underbrace{Q}_{n \times d} \underbrace{e_j}_{d \times 1}$          ▷ $\Delta_M$ only have entries in $i$-th column
6:                                        ▷ Here $\circ$ denotes entry-wise product
7:      $(\Delta_A)_{*,i} \leftarrow (A_{*,i} \circ (\exp((\Delta_M)_{*,i}) - \mathbf{1}_n))$
8:      $\widetilde{M} \leftarrow M + (\Delta_M)_{*,i} e_i^\top$          ▷ We only update $i$-th column of $M$
9:      $\widetilde{A} \leftarrow A + (\Delta_A)_{*,i} e_i^\top$          ▷ We only update $i$-th column of $A$
10:     Obtain diagonal vector $D_{\mathrm{tmp}}$ from $\mathrm{List}_D[\mathrm{ct}_K - 1].$GETB       ▷ It takes $O(n)$ time
11:     $\widetilde{D} \leftarrow D_{\mathrm{tmp}}^{-1} + \mathrm{diag}(\Delta_A)_{*,i}$
12:     **for** $j = 1 \to n$ **do**
13:         $(\Delta_D)_{j,j} \leftarrow (D_{\mathrm{tmp}})_{j,j}^{-1} - \widetilde{D}_{j,j}^{-1}$
14:     **end for**
15:     **if** $\mathrm{ct}_K < n^a$ **then**
16:         $\mathrm{List}_C[\mathrm{ct}_K - 1].(a, b) \leftarrow ((\Delta_A)_{*,i} \in \mathbb{R}^n, V^\top e_i \in \mathbb{R}^d)$
17:         $\mathrm{List}_D[\mathrm{ct}_K - 1].(a, b) \leftarrow (\Delta_D \in \mathbb{R}^{n \times n}, \widetilde{D}^{-1} \in \mathbb{R}^{n \times n})$     ▷ Diagonal matrices
18:     **else**                               ▷ $\mathcal{T}_{\mathrm{mat}}(n, n^a, d) = n^{\omega(1,1,a)}$ time
19:         RECOMPUTE()          ▷ Algorithm 5. Re-compute everything
20:     **end if**
21:     /\*Referesh the memory\*/
22:     $K \leftarrow \widetilde{K}$
23:     $A \leftarrow \widetilde{A}$
24:     $M \leftarrow \widetilde{M}$
25: **end procedure**
26: **end data structure**

---

**Algorithm 3**

---

1: **data structure** DYNAMICATTENTION          ▷ Theorem B.1
2: **procedure** UPDATEV($i \in [n], j \in [d], \delta$)          ▷ Lemma B.5
3:      $\mathrm{ct}_V \leftarrow \mathrm{ct}_V + 1$
4:      **if** $\mathrm{ct}_V < n^a$ **then**
5:         $\mathrm{List}_V[\mathrm{ct}_V - 1].(a, b) \leftarrow (e_i \in \mathbb{R}^n, \delta e_j \in \mathbb{R}^d)$
6:      **else**
7:         RECOMPUTE()          ▷ Algorithm 5. Re-compute everything
8:      **end if**
9: **end procedure**
10: **end data structure**

---

## B.1    INITIALIZATION

We first give the running time of the initialization procedure.

**Lemma B.3** (Init). *The procedure* INIT *(Algorithm 1) takes* $\mathcal{T}_{\mathrm{mat}}(n, d, n)$ *time.*

*Proof.* It is trivially from applying fast matrix multiplication.      □

## B.2    UPDATE

Next, we give the running time of updating $K$.

**Lemma B.4** (Running time of UPDATEK). *The procedure* UPDATEK *(Algorithm 2) takes*

- *Part 1.* $\mathcal{T}_{\mathrm{mat}}(n, n, n^a)$ *time in the worst case*

---

**Algorithm 4** Algorithm that query the $\{i,j\}$-th element in the target matrix

---

1: **data structure** DYNAMICATTENTION             ▷ Theorem B.1
2: **procedure** QUERY($i \in [n], j \in [d]$)          ▷ Lemma B.7, B.6
3:      Let $\Delta_{V,1}$ and $\Delta_{V,2}$ be rectangular matrix obtained from list from $V$
4:      Let $(D_{\mathrm{tmp}})_i^{-1}$ denote the list of diagonal matrices obtained from $\mathrm{List}_D[\mathrm{ct}_K].\mathrm{GETB}$   ▷ This takes $O(1)$ time
5:      /*Below is the target*/
6:      answer $\leftarrow ((D_{\mathrm{tmp}}^{-1}) \cdot (A) \cdot (V + \Delta_{V,1}\Delta_{V,2}))_{i,j}$
7:      /*The actual computation*/
8:      /*Part 1. Answer, This is fast because we store $C = AV$*/
9:      answer$_1 \leftarrow (D_{\mathrm{tmp}})_i^{-1}(C_{i,j} + (\Delta_{C,1}\Delta_{C,2})_{i,j})$          ▷ $O(n^a)$ time
10:      /*Part 2. Answer, this is fast because each column of $\Delta_{V,1}$ and row of $\Delta_{V,2}$ is 1-sparse*/
11:      answer$_2 \leftarrow (D_{\mathrm{tmp}})_i^{-1} A_{i,*} \Delta_{V,1}(\Delta_{V,2})_{*,j}$        ▷ $O(n^a)$ time
12:      answer $\leftarrow \sum_{j=1}^{2}$ answer$_j$
13:      **return** answer
14: **end procedure**
15: **end data structure**

---

**Algorithm 5** Algorithm that re-compute evreything

---

1: **data structure** DYNAMICATTENTION             ▷ Theorem B.1
2: **procedure** RECOMPUTE()           ▷ Lemma B.9, Lemma B.8
3:      Let $\Delta_{C,1}$ and $\Delta_{C,2}$ be rectangular matrix obtained from $\mathrm{List}_C$
4:      Let $\Delta_{V,1}$ and $\Delta_{V,2}$ be rectangular matrix obtained from $\mathrm{List}_V$
5:      Let $\Delta_D(i)$ denote the list of diagonal matrices obtained from $\mathrm{List}_D[i].\mathrm{GETA}$
6:      $\widetilde{C} \leftarrow C + \Delta_{C,1} \cdot \Delta_{C,2} + A\Delta_{V,1} \cdot \Delta_{V,2}$      ▷ It takes $\mathcal{T}_{\mathrm{mat}}(n, n^a, d)$ time
7:      $\widetilde{V} \leftarrow V + \Delta_{V,1} \cdot \Delta_{V,2}$          ▷ It takes $\mathcal{T}_{\mathrm{mat}}(n, n^a, d)$ time
8:      $\Delta_D \leftarrow \sum_{i=1}^{\mathrm{ct}_K} \Delta_D(i)$           ▷ It takes $n^{1+a}$ time
9:      $\widetilde{D}^{-1} \leftarrow D^{-1} + \Delta_D$             ▷ It takes $n$ time
10:      $\widetilde{B} \leftarrow \widetilde{D}^{-1} \cdot \widetilde{C}$             ▷ This takes $nd$
11:      /*Refresh the memory*/
12:      $D \leftarrow \widetilde{D}, C \leftarrow \widetilde{C}, B \leftarrow \widetilde{B}, V \leftarrow \widetilde{V}$
13:      /*Reset the counter*/
14:      $\mathrm{ct}_K \leftarrow 0, \mathrm{ct}_V \leftarrow 0$
15: **end procedure**
16: **end data structure**

---

- *Part 2. $\mathcal{T}_{\mathrm{mat}}(n, n, n^a)/n^a$ time in the amortized case*

*Proof.* **Part 1.** It trivially from Lemma B.9

**Part 2.** If the $\mathrm{ct}_K < n^a$, we pay $O(n)$ time. If $\mathrm{ct}_K = n^a$, we pay $n^{\omega(1,1,a)}$. So the amortized time is

$$\frac{n(n^a - 1) + n^{\omega(1,1,a)}}{n^a} = O(n^{\omega(1,1,a)-a})$$

Note that, by using fast matrix multiplication and the fact that $d = O(n)$, we have $n^{\omega(1,1,a)} = \mathcal{T}_{\mathrm{mat}}(n, n^a, d)$. Thus we complete the proof.      □

Now, we give the running time of updating $V$.

**Lemma B.5** (Running time of UPDATEV). *The procedure* UPDATEV *(Algorithm 3) takes*

- *Part 1. $\mathcal{T}_{\mathrm{mat}}(n, n, n^a)$ time in the worst case.*

- *Part 2. $\mathcal{T}_{\mathrm{mat}}(n, n, n^a)/n^a$ time in the amortized case.*

*Proof.* **Part 1.** It trivially from Lemma B.9.

**Part 2.** If the $\text{ct}_K < n^a$, we pay $O(n)$ time. If $\text{ct}_K = n^a$, we pay $n^{\omega(1,1,a)}$. So the amortized time is

$$\frac{n(n^a - 1) + n^{\omega(1,1,a)}}{n^a} = O(n^{\omega(1,1,a)-a})$$

Note that, by using fast matrix multiplication and the fact that $d = O(n)$, we have $n^{\omega(1,1,a)} = \mathcal{T}_{\text{mat}}(n, n^a, d)$. Thus we complete the proof. $\qquad\square$

### B.3 QUERY

We show the correctness of our QUERY that queries only one element in the target matrix.

**Lemma B.6** (Correctness of QUERY). *The procedure* QUERY *(Algorithm 4) outputs*

$$\begin{aligned}
\widetilde{B}_{i,j} &= (D^{-1} \cdot A \cdot (V + \Delta_V))_{i,j} \\
&= (D^{-1}AV + D^{-1}A\Delta_V)_{i,j}
\end{aligned}$$

*Proof.* Let $\Delta_{V,1}$ denote the vector obtained from $\text{List}_D[\text{ct}_K].\text{GETA}$.

Let $\Delta_{V,2}$ denote the vector obtained from $\text{List}_D[\text{ct}_K].\text{GETB}$

Let $(D_{\text{tmp}})_i^{-1}$ denote the list of diagonal matrices obtained from $\text{List}_D[\text{ct}_K].\text{GETB}$

We know

$$\begin{aligned}
\widetilde{B} &= ((D_{\text{tmp}}^{-1}) \cdot (A) \cdot (V + \Delta_{V,1}\Delta_{V,2})) \\
&= (D_{\text{tmp}})^{-1}(AV) + (D_{\text{tmp}})^{-1}(A\Delta_{V,1}\Delta_{V,2})
\end{aligned}$$

For the $\{i, j\}$-th element, by using simple algebra, we have

$$\begin{aligned}
\widetilde{B}_{i,j} &= (D_{\text{tmp}})_i^{-1}(AV)_{i,j} + (D_{\text{tmp}})_i^{-1}(A\Delta_{V,1}\Delta_{V,2}) \\
&= (D_{\text{tmp}})_i^{-1}(C + \Delta_{C,1} \cdot \Delta_{C,2})_{i,j} + (D_{\text{tmp}})_i^{-1}(A\Delta_{V,1}\Delta_{V,2})_{i,j} \\
&= (D_{\text{tmp}})_i^{-1}(C + \Delta_{C,1} \cdot \Delta_{C,2})_{i,j} + (D_{\text{tmp}})_i^{-1}A_{i,*}\Delta_{V,1}(\Delta_{V,2})_{*,j}
\end{aligned}$$

We know

$$\text{answer}_1 = (D_{\text{tmp}})_i^{-1}(C + \Delta_{C,1} \cdot \Delta_{C,2})_{i,j}$$

and

$$\text{answer}_2 = (D_{\text{tmp}})_i^{-1}A_{i,*}\Delta_{V,1}(\Delta_{V,2})_{*,j}$$

By summing up $\text{answer}_1$ and $\text{answer}_2$, we have

$$\widetilde{B}_{i,j} = (D^{-1}AV + D^{-1}A\Delta_V)_{i,j}.$$

Now, we complete the proof. $\qquad\square$

Next, we give the running time of it.

**Lemma B.7** (Running time of QUERY). *The running time of procedure* QUERY *(Algorithm 4) is* $O(n^a)$.

*Proof.* We first stack all the vectors in $\text{List}_V$ to $\Delta_{V,1} \in \mathbb{R}^{n \times n^a}$ and $\Delta_{V,2} \in \mathbb{R}^{n^a \times d}$, which takes $O(1)$ time.

- Computing $(D_{\text{tmp}})_i^{-1}(C + \Delta_{C,1} \cdot \Delta_{C,2})_{i,j}$ takes $O(n^a)$ time.

- Computing $(\Delta_{V,1}\Delta_{V,2})$ takes $O(n^a)$ time as $\Delta_{V,1}$ is 1-sparse in columns and $(\Delta_{V,2})$ is 1-sparse in rows.

- Computing $(D_{\text{tmp}})_i^{-1}A_{i,*}(\Delta_{V,1}\Delta_{V,2})_{*,j}$ takes $O(n^a)$ time as $\text{nnz}((\Delta_{V,1}\Delta_{V,2})_{*,j}) \leq n^a$.

Hence, the total running time needed is $O(n^a)$ $\qquad\square$

### B.4 RE-COMPUTE

We show the correctness of our re-compute function.

**Lemma B.8** (Correctness of RECOMPUTE). *The procedure* RECOMPUTE *(Algorithm 5) correctly re-compute $D, C, B, V$.*

*Proof.* **Part 1.** Re-compute $D$

Let $\Delta_D(i)$ denote the list of diagonal matrices obtained from $\mathrm{List}_D[i].\mathrm{GETA}$. Then, the total difference between the updated $\widetilde{D}$ and $D$ is $\sum_{i=1}^{\mathrm{ct}_K} \Delta_D(i)$.

By computing $\widetilde{D}^{-1} \leftarrow D^{-1} + \Delta_D$, we correctly get the updated $\widetilde{D}^{-1}$. By computing the inverse of a diagonal matrix we get $\widetilde{D}$.

**Part 2.** Re-compute $V$

We first stack all the vectors in $\mathrm{List}_V$ to $\Delta_{V,1} \in \mathbb{R}^{n \times n^a}$ and $\Delta_{V,2} \in \mathbb{R}^{n^a \times d}$.

By using Fact A.1, we have $\widetilde{V} = V + \Delta_{V,1} \cdot \Delta_{V,2}$.

**Part 3.** Re-compute $C$

Similar to the proof of re-computing $V$.

We first stack all the vectors in $\mathrm{List}_C$ to $\Delta_{C,1} \in \mathbb{R}^{n \times n^a}$ and $\Delta_{C,2} \in \mathbb{R}^{n^a \times d}$.

By using Fact A.1, we have $\widetilde{C} = C + \Delta_{C,1} \cdot \Delta_{C,2} + A\Delta_{V,1} \cdot \Delta_{V,2}$.

**Part 4.** Re-compute $B$

By using the definition of $B = D^{-1}C$, we can update $B$ by using $\widetilde{B} = \widetilde{D}^{-1} \cdot \widetilde{C}$.

Now, we complete the proof. $\square$

Next, we give the running time of it.

**Lemma B.9** (Running time of RECOMPUTE). *The running time of procedure* RECOMPUTE *(Algorithm 5) is $\mathcal{T}_{\mathrm{mat}}(n, n^a, d)$.*

*Proof.* We first stack all the vectors in $\mathrm{List}_V$ to $\Delta_{V,1} \in \mathbb{R}^{n \times n^a}$ and $\Delta_{V,2} \in \mathbb{R}^{n^a \times d}$, which takes $O(1)$ time.

We stack all the vectors in $\mathrm{List}_C$ to $\Delta_{C,1} \in \mathbb{R}^{n \times n^a}$ and $\Delta_{C,2} \in \mathbb{R}^{n^a \times d}$, which takes $O(1)$ time.

- Computing $C + \Delta_{C,1} \cdot \Delta_{C,2} + A\Delta_{V,1} \cdot \Delta_{V,2}$ takes $\mathcal{T}_{\mathrm{mat}}(n, n^a, d)$ time.

- Computing $V + \Delta_{V,1} \cdot \Delta_{V,2}$ takes $\mathcal{T}_{\mathrm{mat}}(n, n^a, d)$ time.

- Computing $\sum_{i=1}^{\mathrm{ct}_K} \Delta_D(i)$ takes $O(n^{a+1})$ time as $\mathrm{nnz}(\Delta_D(i)) = O(n)$ and $\mathrm{ct}_K = O(n^a)$.

- Computing $D^{-1} + \Delta_D$ takes $O(n)$ time as $\mathrm{nnz}(\Delta_D) = O(n)$.

- Computing $\widetilde{D}^{-1} \cdot \widetilde{C}$ takes $O(nd)$ time as $\widetilde{D}^{-1}$ is a diagonal matrix. Hence, the total running time is $\mathcal{T}_{\mathrm{mat}}(n, n^a, d)$.

$\square$

## C  MAIN LOWER BOUND

In Section C.1, we give the definition of Online Matrix Vector (OMV) problem. In Section C.2, we introduce the definition of Hinted MV and its conjecture (from previous work Brand et al. (2019)). In Section C.3, we show the hardness of computing the target matrix with the normalization factor.

### C.1 ONLINE MATRIX VECTOR MULTIPLICATION

Before studying the hardness of our problem, we first review a famous problem in theoretical computer science which is called online matrix vector multiplication problem. Here is the definition of online matrix vector multiplication, which has been a crucial task in many fundamental optimization problems.

**Definition C.1** (Online Matrix Vector (OMV) Henzinger et al. (2015); Larsen & Williams (2017); Chakraborty et al. (2018))**.** *Given a matrix $A \in \{0,1\}^{n \times n}$, let $T = O(n)$, there is an online sequence of vectors $u_1, \cdots, u_T \in \{0,1\}^n$. The goal is to design a structure that whenever receives a new vector $u_t$ and output $Au_t$.*

Such a problem is widely believed in the community that there is no algorithm to solve it in truly subquadratic time per vector and there is no algorithm to solve it in truly subcubic time over all vectors.

### C.2 HARDNESS FROM PREVIOUS WORK

We define the hinted Mv problem from previous work Brand et al. (2019).

**Definition C.2** (Hinted MV (HMV) (Brand et al., 2019, Definition 5.6))**.** *Let the computations be performed over the boolean semi-ring and let $m = n^\tau$, $0 < \tau \leq 1$. The hinted $Mv$ problem consists of the following phases:*

1. *Input two $n \times n$ matrices $M$ and $V$*

2. *Input an $n \times n$ matrix $P$ with at most $n^\tau$ non-zero entries*

3. *Input a single index $i \in [n]$*
   - *We need to answer $MPV_{*,i}$*
   - *Here $V_{*,i} \in \mathbb{R}^n$ is the $i$-th column of matrix $V$*

We give the hinted Mv conjecture which is from prior work Brand et al. (2019).

**Conjecture C.3** (HMV conjecture (Brand et al., 2019, Conjecture 5.2), restatement of Conjecture 3.1)**.** *For every constant $0 < \tau \leq 1$ no algorithm for the hinted $Mv$ problem (Definition C.2) can simultaneously satisfy*

- *polynomial time in phase 1*

- *$O(n^{\omega(1,1,\tau)-\epsilon})$ time complexity in phase 2 and*

- *$O(n^{1+\tau-\epsilon})$ in phase 3*

*for some constant $\epsilon > 0$.*

### C.3 ONLINE DIAGONAL-NORMALIZED ATTENTION MATRIX VECTOR MULTIPLICATION

Next, we consider the normalization factor and defined the problem as the following.

**Definition C.4** (ODAMV$(n,d)$, restatement of Definition 1.2)**.** *The goal of **O**nline **D**iagonal-based normalized **A**ttention **M**atrix **V**ector Multiplication problem ODAMV$(n,d)$ is to design a data structure that satisfies the following operations:*

1. INIT*: Initialize on $n \times d$ matrices $Q$, $K$, $V$.*

2. UPDATE*: Change any entry of $Q$, $K$, or $V$.*

3. QUERY*: For any given $i \in [n]$, $j \in [d]$, return $(D^{-1}\exp(QK^\top)V)_{i,j}$, where $D = \mathrm{diag}(\exp(QK^\top)\mathbf{1}_n)$.*

Next, we present our lower bound result with the normalization factor.

**Lemma C.5.** *Assuming the hinted $Mv$ conjecture (Conjecture C.3): For every constant $0 < \tau \le 1$, there is no algorithm that solve $\mathsf{ODAMV}(n, d)$ problem (Definition C.4) with*

- *polynomial initialization time, and*

- *amortized update time $O(\mathcal{T}_{\mathrm{mat}}(n, n^\tau, d)/n^{\tau + \Omega(1)})$, and*

- *worst query time $O(n^{\tau - \Omega(1)})$.*

*Proof.* Assume there was a dynamic algorithm faster than what is stated in Lemma C.5 for some parameter $\tau$, i.e. update time $O(\mathcal{T}_{\mathrm{mat}}(n, n^\tau, d)/n^{\tau + \epsilon})$ and query time $O(n^{\tau - \epsilon})$ for some constant $\epsilon > 0$. We show that this would contradict the hinted $Mv$ conjecture (Conjecture C.3).

Let us take an instance for the $v$-hinted Mv problem (Definition C.2) with $M \in \{0, 1\}^{n \times n}, V \in \{0, 1\}^{n \times n}$.

We can construct matrix $\mathsf{M} \in \{0, 1\}^{n \times 2n}$ and $\mathsf{V} \in \{0, 1\}^{2n \times n}$ as follows

$$\mathsf{M} := \begin{bmatrix} M & \overline{M} \end{bmatrix} \quad \text{and} \quad \mathsf{V} := \begin{bmatrix} V \\ \mathbf{0}_{n \times n} \end{bmatrix}$$

where $\overline{M}$ is a matrix that $\overline{M}_{i,j} = 1 - M_{i,j}$.

Note that $\|\mathsf{M}_{i,*}\|_1 = n$, for each $i \in [n]$.

Based on the above construction, we will create a new instance $\mathsf{ODAMV}(\widetilde{n} = 2n, \widetilde{d} = 2n)$, where

$$\widetilde{Q} = \begin{bmatrix} \mathsf{M} \\ \mathbf{0}_{n \times 2n} \end{bmatrix}, \quad \widetilde{K} = \mathbf{0}_{2n \times 2n}, \quad \widetilde{V} = \begin{bmatrix} \mathsf{V} & \mathbf{0}_{2n \times n} \end{bmatrix}$$

During phase 1, we give this input to the dynamic algorithm for the $\mathsf{ODAMV}$ problem (Definition C.4).

Let $D \in \{0, 1\}^{n \times n}$ denote a diagonal matrix, where $\mathrm{nnz}(D) = n^\tau$

During phase 2, we receive the $2n \times 2n$ diagonal matrix $\mathsf{P}$, where

$$\mathsf{P} = \begin{bmatrix} P & 0 \\ 0 & P \end{bmatrix}$$

and $\mathrm{nnz}(\mathsf{P}) = 2n^\tau$.

We perform $2n^\tau$ updates to the data structure to set $\widetilde{K}^\top = \mathsf{P}$. This takes

$$O(\widetilde{n}^\tau \cdot (\mathcal{T}_{\mathrm{mat}}(\widetilde{n}, \widetilde{n}^\tau, \widetilde{d})/\widetilde{n}^{\tau + \epsilon})) = O(n^{\omega(1,1,\tau) - \epsilon})$$

time.

Note that

- $\|\widetilde{Q}_{i,*}\|_1 = n$, for each $i \in [n]$.

- $\|\widetilde{Q}_{i,*}\|_1 = 0$, for each $i \in [n+1, 2n]$.

By using the definition of $\mathsf{P}$, we know that, for each $i \in [n]$

$$\widetilde{D}_{i,i} = n^\tau \exp(1) + n^\tau \exp(0) = n^\tau (e + 1).$$

For each $i \in [n+1, 2n]$

$$\widetilde{D}_{i,i} = n^\tau \exp(0) = n^\tau. \tag{2}$$

Hence, we don't need to update $\widetilde{D}$.

At last, in phase 3, we perform $\widetilde{n}$ queries to obtain the column $\exp(\widetilde{Q}\widetilde{K}^\top)\widetilde{V}_{*,i}$ in $O(\widetilde{n} \cdot \widetilde{n}^{\tau - \epsilon}) = O(n^{1 + \tau - \epsilon})$ time.

Using Claim C.7 and Claim C.6, we know that, for any $i \in [n]$ and for any $j \in [n]$, if there is an algorithm that can find $(\widetilde{D}^{-1} \exp(\widetilde{Q}\widetilde{K}^{\top})\widetilde{V})_{j,i}$, then using $(\widetilde{D}^{-1} \exp(\widetilde{Q}\widetilde{K}^{\top})\widetilde{V})_{j,i} - (\widetilde{D}^{-1}\widetilde{V})_{j,i}$ is enough to reconstruct $(\mathsf{MPV})_{j,i}$. Here $\widetilde{D}^{-1}\widetilde{V}$ can be computed in just $O(1)$ time via Eq. (2). Thus, we can know the $(MDV)_{j,i}$ for the hinted $Mv$ problem in $O(n^{1+\tau\epsilon})$ time, contradicting the hinted $Mv$ conjecture.

$\square$

**Claim C.6.** *For each $i \in [n]$ and $j \in [n]$, if $(\widetilde{D}^{-1}(\exp(\widetilde{Q}\widetilde{K}^{\top}) - \mathbf{1}_{\widetilde{n}\times\widetilde{n}})\widetilde{V})_{j,i}$ is $> 0$, then* $(\mathsf{MPV})_{j,i} = 1$,

*Proof.* By using the fact that $n^{\tau}(e+1) > 0$ and $n^{\tau} > 0$, we have
$$\widetilde{D}^{-1}(\exp(\widetilde{Q}\widetilde{K}^{\top}) - \mathbf{1}_{\widetilde{n}\times\widetilde{n}})\widetilde{V})_{j,i} > 0$$
$$((\exp(\widetilde{Q}\widetilde{K}^{\top}) - \mathbf{1}_{\widetilde{n}\times\widetilde{n}})\widetilde{V})_{j,i} > 0$$

We know
$$\widetilde{Q} = \begin{bmatrix} \mathsf{M} \\ \mathbf{0}_{n\times 2n} \end{bmatrix}, \quad \widetilde{K}^{\top} = \begin{bmatrix} P & 0 \\ 0 & P \end{bmatrix}, \quad \widetilde{V} = \begin{bmatrix} \mathsf{V} & \mathbf{0}_{2n\times n} \end{bmatrix},$$

so we have
$$((\exp(\mathsf{MP}) - \mathbf{1}_{n\times 2n})\mathsf{V})_{j,i} > 0.$$

For $k \in [n+1, 2n]$, as $\mathsf{V} = \begin{bmatrix} V \\ \mathbf{0}_{n\times n} \end{bmatrix}$, we know $(\exp(\mathsf{MP}) - \mathbf{1}_{n\times 2n})_{j,k}(\mathsf{V})_{k,i} = 0$.

Using the definition of matrix multiplication, and the fact that $\exp(x) > 1$ for all $x > 0$, we have some $k \in [n]$ with
$$(\exp(\mathsf{MP}) - \mathbf{1}_{n\times 2n})_{j,k}(\mathsf{V})_{k,i} > 0$$
$$(\exp(\mathsf{MP})_{j,k} - 1)(\mathsf{V})_{k,i} > 0$$
We can conclude that for each $i \in [n], j \in [n]$, there is at least one $k \in [n]$ such that

- $\mathsf{V}_{k,i} > 0$

- $(\mathsf{MP})_{j,k} > 0$

Therefore, by using the definition of boolean semi-ring, we can conclude that $(\mathsf{MPV})_{j,i} = 1$

$\square$

**Claim C.7.** *For each $i \in [n]$ and $j \in [n]$, if $(\widetilde{D}^{-1}(\exp(\widetilde{Q}\widetilde{K}^{\top}) - \mathbf{1}_{\widetilde{n}\times\widetilde{n}})\widetilde{V})_{j,i}$ is $0$ then $(\mathsf{MPV})_{j,i} = 0$.*

*Proof.* By using the fact that $n^{\tau}(e+1) > 0$ and $n^{\tau} > 0$, we have
$$\widetilde{D}^{-1}(\exp(\widetilde{Q}\widetilde{K}^{\top}) - \mathbf{1}_{\widetilde{n}\times\widetilde{n}})\widetilde{V})_{j,i} = 0$$
$$((\exp(\widetilde{Q}\widetilde{K}^{\top}) - \mathbf{1}_{\widetilde{n}\times\widetilde{n}})\widetilde{V})_{j,i} = 0$$

We know
$$\widetilde{Q} = \begin{bmatrix} \mathsf{M} \\ \mathbf{0}_{n\times 2n} \end{bmatrix}, \quad \widetilde{K}^{\top} = \begin{bmatrix} P & 0 \\ 0 & P \end{bmatrix}, \quad \widetilde{V} = \begin{bmatrix} \mathsf{V} & \mathbf{0}_{2n\times n} \end{bmatrix},$$

so we have
$$((\exp(\mathsf{MP}) - \mathbf{1}_{n\times 2n})\mathsf{V})_{j,i} = 0.$$

For $k \in [n+1, 2n]$, as $\mathsf{V} = \begin{bmatrix} V \\ \mathbf{0}_{n\times n} \end{bmatrix}$, we know $(\exp(\mathsf{MP}) - \mathbf{1}_{n\times 2n})_{j,k}(\mathsf{V})_{k,i} = 0$.

For all $k \in [n]$ such that $\mathsf{V}_{k,i} = 1$, we have $(\exp(\mathsf{MP}) - \mathbf{1}_{n\times 2n})_{j,k} = 0$, which also implies that $(\mathsf{MP})_{j,k} = 0$.

Now, we can conclude that $(\mathsf{MPV})_{j,i} = 0$ for each $i \in [n]$ and $j \in [n]$. $\square$