# OpenReview forum: "Algorithm and Hardness for Dynamic Attention Maintenance in Large Language Models"
_ICLR.cc/2024/Conference — Submitted to ICLR 2024_

### Official Review · Reviewer_wrST · 2023-10-31

**Soundness:** 3 good
**Presentation:** 3 good
**Contribution:** 3 good
**Rating:** 8
**Confidence:** 3

**Summary:**

This paper addresses the computation of attention matrix in a dynamic setting where an entry of one of matrices corresponding to query and key tokens can make a change. In static case where input matrices are fixed, it is known that attention computation cannot be performed in subquadratic time when input matrices have larger entries, even if an additive approximation error is permitted. This paper proposes an algorithm for the online version such that an update takes only amortized subquadratic time and a query, computing one entry of attention, takes only worst-case subquadratic time, without approximation. This paper also suggests a conditional lower bound for the time complexity of online attention matrix computation that matches the complexity of the proposed algorithm.

**Strengths:**

The finest contribution I think is that it achieves truly subquadratic ($O(n^{2-\Omega(1)})$) time for attention matrix computation without approximation by considering dynamic setting. As shown in [Alman & Song, 2023], in the static setting where each matrix is given and fixed in advance, the attention matrix computation cannot be performed in $O(n^{2-\Omega(1)})$ time when the entries of matrices are not bounded between $-\sqrt{\log n}$ and $\sqrt{\log n}$ even an additive error of $1/\mathrm{poly}(n)$ is permitted. This paper considers dynamic setting where an update of one entry of matrix and a query for computing one entry of attention come in an online manner. This paper proposes a data structure satisfying both subquadratic update time and sublinear query time.
Here I mention the query time; since the query in dynamic setting only computes one entry of the result matrix, one suspects that the full attention computation might take more than quadratic time. However, the result of [Alman & Song, 2023] holds even when the number of columns, $d$, of input matrices is far smaller than $n$, i.e., $d=O(\log n)$. In this case, since there are only $nd=O(n\log n)$ entries for the result matrix, the full computation of attention takes only subquadratic time. Note that this does not violate the result of [Alman & Song, 2023] since the proposed algorithm takes at least quadratic time for preprocessing.

Moreover, this paper also proves a matching conditional lower bound for the time complexity, meaning that the proposed algorithm is optimal if the conjecture is true.

**Weaknesses:**

* The online setting dealt in this paper has less reality; in the dynamic setting of this problem, only one entry of input matrices is changed by one update. However, in real situation, although it can be assumed that the values of entries change slowly, a substantial portion of entries are subject to change. Therefore, when applied to real situations, polynomially many update operations may occur, which eventually result in quadratic computation time.
* The proposed algorithm relies heavily on fast matrix multiplication. Therefore, I think the proposed algorithm is practically useless although its theoretical implication is inevitable.

**Questions:**

* As described in "weaknesses" section, I think this work has of theoretical interest, and so the worst-case update time instead of amortized update time is preferable if possible. As described in Remark B.2, since the proposed algorithm relies on the periodic rebuilding of data structures, we can make the update time fit for worst-case analysis theoretically by dividing the rebuilding of data structures into updates. So, the question is, why the main result (Theorem 1.3 or Theorem B.1) is posed on amortized update time? For clarity?
* I'm interested in the situation that we are in a static setting but we only have to compute one entry of attention matrix. In such a case, there is possibility to lower time complexity, but it is not straightforward because it involves $n\times n$ matrix's inverse. Do you have any idea for this? If so, it complements this work since in the dynamic setting we also have to compute only one entry of attention for a query.

---

> ### Author Response · Authors · 2023-11-21
> **Rebuttal by Authors - Part 1**
>
> **Q1**: The online setting dealt in this paper has less reality; in the dynamic setting of this problem, only one entry of input matrices is changed by one update. However, in real situation, although it can be assumed that the values of entries change slowly, a substantial portion of entries are subject to change. Therefore, when applied to real situations, polynomially many update operations may occur, which eventually result in quadratic computation time.
>
> **A1**: Your careful reading and the insightful suggestions are greatly appreciated. In response to the concerns raised regarding the practical implications of the dynamic attention maintenance problem, we emphasise that entry-wise update of the weight matrix is prevalent in real-world scenarios. Deep neural network architectures frequently exhibit significant redundancy, and empirical evidence supports the capacity of deep neural networks to tolerate substantial levels of sparsity [1,2]. In downstream fine-tuning tasks, the dimensions of the model often make the fine tuning infeasible.
> Sparse neural networks have demonstrated enhanced parameter and computational efficiency relative to dense networks, and, in many cases, can significantly decrease wall clock inference times [3,4]. Over the past few years, numerous techniques for inducing sparsity have been proposed to sparsify the neural network such as magnitude pruning [1,5], RegL [6] and dynamic sparse reparameterization [7]. For attention based models, leveraging the sparsity of the attention can help make the model more efficient, as in [8,9,10].
> For example, to save the computation time and memory of the transformer, [11] only computes the top-k entries for each row in the attention matrix. Their work implies that we do not need to update the entire query and key matrix. A promising avenue for future exploration involves the adaptation of this concept to implement rank-1 updates, potentially surpassing entry-wise updates in effectiveness. We believe that it is an interesting direction for enhancing practical training efficiency. Our approach acknowledges the value of simplicity as an initial foundation, drawing from a range of theoretical papers [12,13,14,15]. Starting with a simpler version allows us to build a solid starting point for understanding the broader context.
>
> [1] Han, Song, et al. "Learning both weights and connections for efficient neural network." Advances in Neural Information Processing Systems 28 (2015).
>
> [2] Gale, Trevor, Erich Elsen, and Sara Hooker. "The state of sparsity in deep neural networks." arXiv preprint arXiv:1902.09574 (2019).
>
> [3] Theis, Lucas, et al. "Faster gaze prediction with dense networks and Fisher pruning." arXiv preprint arXiv:1801.05787 (2018).
>
> [4] Kalchbrenner, Nal, et al. "Efficient neural audio synthesis." International Conference on Machine Learning. PMLR, 2018.
>
> [5] Zhu, Michael, and Suyog Gupta. "To prune, or not to prune: exploring the efficacy of pruning for model compression." arXiv preprint arXiv:1710.01878 (2017).
>
> [6] Evci, Utku, et al. "Rigging the lottery: Making all tickets winners." International Conference on Machine Learning. PMLR, 2020.
>
> [7] Mostafa, Hesham, and Xin Wang. "Parameter efficient training of deep convolutional neural networks by dynamic sparse reparameterization." International Conference on Machine Learning. PMLR, 2019.
>
> [8] Roy, Aurko, et al. "Efficient content-based sparse attention with routing transformers." Transactions of the Association for Computational Linguistics 9 (2021): 53-68.
>
> [9] Choromanski, Krzysztof, et al. "Rethinking attention with performers." arXiv preprint arXiv:2009.14794 (2020).
>
> [10] Kaiser, Łukasz, and Samy Bengio. "Discrete autoencoders for sequence models." arXiv preprint arXiv:1801.09797 (2018).
>
> [11] Gupta, Ankit, et al. "Memory-efficient Transformers via Top-$ k $ Attention." arXiv preprint arXiv:2106.06899 (2021).
>
> [12] Alman, Josh, and Zhao Song. "Fast attention requires bounded entries." arXiv preprint arXiv:2302.13214 (2023).
>
> [13] Deng, Yichuan, Sridhar Mahadevan, and Zhao Song. "Randomized and deterministic attention sparsification algorithms for over-parameterized feature dimension." arXiv preprint arXiv:2304.04397 (2023).
>
> [14] Kitaev, Nikita, Łukasz Kaiser, and Anselm Levskaya. "Reformer: The efficient transformer." arXiv preprint arXiv:2001.04451 (2020).
>
> [15] Katharopoulos, Angelos, et al. "Transformers are rnns: Fast autoregressive transformers with linear attention." International conference on machine learning. PMLR, 2020.

---

> ### Author Response · Authors · 2023-11-22
> **Rebuttal by Authors - Part 2**
>
> **Q2**: The proposed algorithm relies heavily on fast matrix multiplication. Therefore, I think the proposed algorithm is practically useless although its theoretical implication is inevitable.
>
> **A2**: We appreciate your concerns regarding the practical applicability of our algorithm given its reliance on fast matrix multiplication. However, it is important to clarify that our algorithm does not strictly require the "fastest" matrix multiplication methods to achieve non-trivial improvements. As discussed in our previous rebuttal, our algorithm is designed to work efficiently even with subcubic-time matrix multiplication algorithms like Strassen's. Specifically, using the Strassen algorithm, our update time is T(n,n,n^a) = n^(2.8075 * a + 2-2a), which is faster than the naïve n^2 update time for all choices of 0<a<1. This demonstrates that our approach can outperform the simple O(n^2) update strategy even without resorting to the most advanced matrix multiplication algorithms. Furthermore, by choosing a=0, our algorithm aligns with the naive n^3 time matrix multiplication, ensuring that it is not slower than the naive approach, with an n^2 update and O(1) query time. Thus, while the theoretical underpinnings of our work indeed leverage advanced concepts in matrix multiplication, the algorithm is designed to be adaptable and practical, even when less advanced multiplication techniques are employed.
>
> **Q3**: As described in "weaknesses" section, I think this work has of theoretical interest, and so the worst-case update time instead of amortized update time is preferable if possible. As described in Remark B.2, since the proposed algorithm relies on the periodic rebuilding of data structures, we can make the update time fit for worst-case analysis theoretically by dividing the rebuilding of data structures into updates. So, the question is, why the main result (Theorem 1.3 or Theorem B.1) is posed on amortized update time? For clarity?
>
> **A3**: Thanks for pointing that out. In our paper, we don’t go into detail for worst-case as the worst case time can be derived from our results by simply using standard techniques as stated in Remard B.2. We believe showing the proof of the worst case would confuse the readers.
>
> **Q4**: I'm interested in the situation that we are in a static setting but we only have to compute one entry of attention matrix. In such a case, there is possibility to lower time complexity, but it is not straightforward because it involves n×n  matrix's inverse. Do you have any idea for this? If so, it complements this work since in the dynamic setting we also have to compute only one entry of attention for a query.
>
> **A4**: One entry can be computed statically in O(nd) time: Compute one row of QK, then do entry-wise exponentiation and compute inner product with column of V. Normalization works because we have the necessary row of exp(QK).

---

> > ### Comment · Reviewer_wrST · 2023-11-23
> >
> > Thank you very much for detailed reply, especially for the reality of the online setting that is somewhat convincing for me. Despite of the other reviewers' concerns, I still consider this work theoretically interesting, and so I will keep the score.

---

### Official Review · Reviewer_ymZ9 · 2023-10-31

**Soundness:** 3 good
**Presentation:** 3 good
**Contribution:** 3 good
**Rating:** 6
**Confidence:** 4

**Summary:**

The paper studies an algorithmic problem that arises in the context of transformers. In transformer architectures, implementing the attention layers corresponds to the following task: Given matrices Q, K, V, compute A = exp(QK^T), and output AV^T with the rows of A scaled by the corresponding row-sums.

Several approaches have been proposed for performing these steps faster by exploiting specific structures of the matrices or by using approximations (e.g., hashing, KDE). However, most work has been for static settings where the matrices Q, K, and V do not change.

The present paper studies a setup where while Q is fixed, K, V can be updated. In each round, one vector of K, V could change and the goal is to still compute the attention mechanism efficiently per each update/query after a pre-processing step. This is an interesting setup; however, a drawback is that Q is not allowed to change, and in most architectures, it would.

The main idea in the current work is to batch together the updates and then use a fast matrix multiplication subroutine to improve on performing each step immediately. The authors also show that their algorithm is optimal under a believable conjecture from algorithms.

**Strengths:**

The paper studies an interesting model that while not capturing the real-world setup exactly could be a useful step. The algorithm proposed is nice and the quantitative bounds are tight.

**Weaknesses:**

The model doesn't allow for Q to be updated which it could be. The algorithm while nice, is a somewhat straightforward adaptation of known ideas.

**Questions:**

Can you address why Q cannot be updated?

---

> ### Author Response · Authors · 2023-11-21
> **Rebuttal by Authors**
>
> **Q1**: The algorithm while nice, is a somewhat straightforward adaptation of known ideas.
>
> **A1**: Thank you for your review. It seems there might be a misunderstanding regarding the novelty and complexity of our algorithm. Our approach is not just a straightforward adaptation of known ideas; it introduces a dynamic data structures for attention computation, a critical component in modern machine learning.
> Firstly, our upper bound result utilizes a 'lazy update strategy,' which significantly improves the efficiency of the attention mechanism. The theorem presented in our work demonstrates that our dynamic data structure efficiently balances space complexity and operational time for updating and querying. Specifically, it allows for a trade-off between update and query time, which is a critical aspect in real-time applications.
> Furthermore, we have achieved better performance than the naive $O(n^2)$ update time, which is a significant improvement. This is true regardless of the fast matrix multiplication algorithm used. For example, with Strassen's algorithm, our update time is $O(n^{2+(1.8075-2)a})$, showcasing the efficiency of our approach.
> Additionally, our second result, which involves the hinted matrix vector multiplication conjecture, provides a lower bound that matches our upper bound. This is crucial as it demonstrates the theoretical optimality of our approach under the given conjecture.
> Our work also contributes to the field of fine-grained complexity theory by identifying the nature of hardness in dynamic attention maintenance. By providing tight upper and conditional lower bounds, we show that this problem belongs to the class of challenges that revolve around the trade-off between matrix-vector multiplication and fast matrix multiplication.
> In summary, while our work builds upon existing ideas, its contribution lies in significantly advancing the efficiency and theoretical understanding of dynamic attention mechanisms.
>
>
>
> **Q2**: Can you address why $Q$ cannot be updated?
>
> **A2**: Thanks for your careful reading. Updating an entry in $Q$, will change an entire row of $A=exp(QK)$. If we store $AV$ implicitly, it will have $O(n)$ query time. If we store it explicitly, it takes $O(nd)$ update time. It would work if we tried to maintain $QKV$ instead of $exp(QK)V$, because then an entry update to $Q$ means we just add a row of $KV$ to the output, which takes $O(n)$ update time. We can maintain $KV$ separately. However, if we want to maintain $exp(QK)V$ as in attention maintenance, then this approach doesn’t work.

---

### Official Review · Reviewer_YepA · 2023-11-01

**Soundness:** 2 fair
**Presentation:** 2 fair
**Contribution:** 2 fair
**Rating:** 3
**Confidence:** 2

**Summary:**

The attention scheme has been widely used in models including LLMs. This paper proposes a dynamic version of the attention matrix multiplicaion problem, in contrast to previous static attention multiplications. Specifically, the paper deal with entry-level update of K and V, as well as entry-level query of the attention matrix. Results in this paper are two folds. The first is an algorithm that supports the dynamic attention with specific amortized update time and worst-case query time. The second is a lower bound indicating that the proposed algorithm will be conditionally optimal given that the hinted matrix vector multiplication conejcture is correct.

**Strengths:**

- The paper provides detailed introduction of dynamic attention matrix multiplication problem, and the hinted MV conjecture.

- A detailed theoretical analysi of the proposed lower bound of the dynamic attention matrix multiplication problem.

**Weaknesses:**

- The paper mentioned attention is used in LLMs, but it seems that the proposed method is not targeted for LLMs.

- Even though this paper is a relatively theoretical paper, it would be better to provide some experiments/simulation results about amortized update time and query time, and comparison with other related works mentioned in the paper.

- The proposed theoretical results cannot cover updating K, V together, which is usually the case for model training/fine-tuning nowadays.

- Minor points:
1. The paper structure is a little bit unbalanced, with the first introduction section taking up more than half of the pages. The related work section (Sec 1.2) can be compressed.

2. The citation style is not in parenthesis (\citep{}).

**Questions:**

- The biggest question I have is why we need dynamic attention multiplication. There are many works for sparse attention and low-rank attention [1, 2], which provide potential solutions for the long-range attention problem. And when we do fine-tuning on LLMs, we can also freeze part of its parameters and only tune the rest. It seems not very common to do targeted update for specific entries of K, V, or query specific positions of the attention matrix. So I'm not sure what is the scenario that the proposed method is mostly suitable for.

[1] Choromanski, Krzysztof, et al. "Rethinking attention with performers." arXiv preprint arXiv:2009.14794 (2020)

[2] Beltagy, Iz, Matthew E. Peters, and Arman Cohan. "Longformer: The long-document transformer." arXiv preprint arXiv:2004.05150 (2020)

**Details Of Ethics Concerns:**

No ethics concerns.

---

> ### Author Response · Authors · 2023-11-21
> **Rebuttal by Authors**
>
> **Q1**: The paper mentioned attention is used in LLMs, but it seems that the proposed method is not targeted for LLMs.
>
> **A1**: Thanks for pointing out that issue. We kindly argue that our method updates the attention matrix in a lazy fashion that speeds up the average update and query time. For LLMs, the attention scheme is the key component that makes it possess the capability to understand and produce complex language. However, the computation of the attention matrix is time consuming. We believe our method is targeting the LLMs to improve the running time.
>
> **Q2**: The paper structure is a little bit unbalanced, with the first introduction section taking up more than half of the pages. The related work section (Sec 1.2) can be compressed.
>
> **A2**: Thanks for pointing it out. To make the structure more balance, we will make our introduction and related work more compressed and also move some of the related work to our appendix. We will be sure to update that in our camera ready version.
>
> **Q3**; The biggest question I have is why we need dynamic attention multiplication. There are many works for sparse attention and low-rank attention [1, 2], which provide potential solutions for the long-range attention problem. And when we do fine-tuning on LLMs, we can also freeze part of its parameters and only tune the rest. It seems not very common to do targeted update for specific entries of $K, V$, or query specific positions of the attention matrix. So I'm not sure what is the scenario that the proposed method is mostly suitable for.
>
> [1] Choromanski, Krzysztof, et al. "Rethinking attention with performers." arXiv preprint arXiv:2009.14794 (2020)
>
> [2] Beltagy, Iz, Matthew E. Peters, and Arman Cohan. "Longformer: The long-document transformer." arXiv preprint arXiv:2004.05150 (2020)
>
> **A3** : The question you've raised about the necessity of dynamic attention multiplication in light of existing sparse and low-rank attention models is indeed pertinent.
> The significance of our approach lies in its ability to leverage the redundancy often found in deep neural networks. As you mentioned, many methods have been proposed to leverage the sparsity to speed up the computation [1,2,3,4,5,6]. For example, in the top-k method [6], that only updates the top-k entries that have the highest attention score. In a lazy fashion, we can accumulate the update of them and recompute the attention after enough iterations. Our proposed method opens the door to future explorations, such as implementing rank-1 updates, which could potentially be more effective than entry-wise updates. It acknowledges the importance of starting with a simpler model as a foundation. This approach not only allows for a clearer understanding of the complex dynamics within LLMs but also provides a solid base for further enhancements in training efficiency, drawing from various theoretical perspectives [7,8,9,10].
>
> [1] Han, Song, et al. "Learning both weights and connections for efficient neural network." Advances in Neural Information Processing Systems 28 (2015).
>
> [2] Gale, Trevor, Erich Elsen, and Sara Hooker. "The state of sparsity in deep neural networks." arXiv preprint arXiv:1902.09574 (2019).
>
> [3] Roy, Aurko, et al. "Efficient content-based sparse attention with routing transformers." Transactions of the Association for Computational Linguistics 9 (2021): 53-68.
>
> [4] Choromanski, Krzysztof, et al. "Rethinking attention with performers." arXiv preprint arXiv:2009.14794 (2020).
>
> [5] Kaiser, Łukasz, and Samy Bengio. "Discrete autoencoders for sequence models." arXiv preprint arXiv:1801.09797 (2018).
>
> [6] Gupta, Ankit, et al. "Memory-efficient Transformers via Top-$ k $ Attention." arXiv preprint arXiv:2106.06899 (2021).
>
> [7] Alman, Josh, and Zhao Song. "Fast attention requires bounded entries." arXiv preprint arXiv:2302.13214 (2023).
>
> [8] Deng, Yichuan, Sridhar Mahadevan, and Zhao Song. "Randomized and deterministic attention sparsification algorithms for over-parameterized feature dimension." arXiv preprint arXiv:2304.04397 (2023).
>
> [9] Kitaev, Nikita, Łukasz Kaiser, and Anselm Levskaya. "Reformer: The efficient transformer." arXiv preprint arXiv:2001.04451 (2020).
>
> [10] Katharopoulos, Angelos, et al. "Transformers are rnns: Fast autoregressive transformers with linear attention." International conference on machine learning. PMLR, 2020

---

### Official Review · Reviewer_tLaa · 2023-11-03

**Soundness:** 3 good
**Presentation:** 2 fair
**Contribution:** 2 fair
**Rating:** 3
**Confidence:** 3

**Summary:**

This work studies the problem of dynamically maintaining an attention matrix (termed ODAMV): given an entry-wise update per iteration to the attention matrix, design an algorithm that can efficiently compute any desired entry of the updated attention matrix upon some query. This work proposes an algorithm based on lazy update techniques with guaranteed amortized run time and worst case run time. Furthermore, this work provides a matching conditional lower bound of the run time of the proposed algorithm, assuming the hardness of Hinted Matrix Vector Multiplication conjecture holds.

**Strengths:**

The proposed problem of dynamically maintaining the attention matrix is novel and interesting.

**Weaknesses:**

$\textbf{Problem definition.}$
During the training or deployment of an LLM, for example, the input sequences (of length $n$) are different per sample. Hence, the matrices $Q, K, V$ are different for each sample, and it is hardly the case that the attention matrix will be updated by only one entry at a time. Is it possible to give a practical scenario where the problem of dynamic/online attention maintenance with entry-wise updates considered in this work applies?

It might be more interesting to consider a row or a column, or even a submatrix of fixed size, being updated in each iteration.

$\textbf{Presentation needs to be greatly improved.}$

- In the abstract, $\omega(1,1,\tau)$ and $\tau$ are undefined notations that make it hard for the readers to understand the core results of the paper at first glance. While one might be able to infer that $\omega$ means the matrix multiplication exponent from the context, it is hard to understand $\omega(1,1,\tau)$. Also, without properly introducing $\tau$, it is unclear whether the run time results hold for a specifically chosen $\tau$ (might be too complicated to write out in the abstract) or for any $\tau\in (0, 1]$.

- In Theorem 1.3, $\delta$ is undefined. I was thinking about some notion of failure probability when I first read this, but it turns out in later sections that $\delta$ is indeed the update to the attention matrix.

- The “Transformer Theory” and the details of classical optimization dynamic matrix maintenance in Section 1.2 “Related Work” do not seem to be directly related to the problem considered in this work. It’d be better to be moved to the Appendix.

- Section 3.1 does not give a precise and clear overview of the lazy update based algorithm that solves ODAMV. For example, it is unclear what “List_C, List_D, List_V” in “Lazy Update” store. In “Fast Query”, it is confusing as why there are two stacked matrices, $\Delta_{V, 1}$ and $\Delta_{V, 2}$ from List_V.

- In Section 3.2 and 4 on the Lower Bound (LB), it is unclear why this work presents the LB result for a simpler problem called OAMV, which is not the true problem ODAMV considered, instead of directly presenting the LB results of ODAMV. Section 4 is essentially a more detailed / formalized version of Section 3.2. These two sections might be combined.

- Minor issue: In Theorem 1.3, “This operation has the same … as K update” should be “This operation has the same … as UpdateK”?

- Minor issue: Below Definition 1.2, “When then complement our result …” => “We then complement our result”

- Minor issue: In the paragraph “Fast Query” in Section 3.1, “denote the lates $D^{-1}$” => “denote the latest $D^{-1}$”

**Questions:**

1. On the algorithm / upper bound side, is there any utility guarantee of the algorithm? For example, given a query to the $(i, j)$-th entry, how close it is between $\widetilde{B}_{i, j}$

and $B_{i, j}$, where $B$ is the target matrix after updates (as defined in Section 3.1) and $\widetilde{B}$ is the updated matrix computed by the algorithm?

2. A follow-up question: why does one need to recompute the attention matrix every $n^{\alpha}$ updates? Is it because after that many updates, certain utility guarantee no longer holds?

3. What is the algorithmic challenge in designing the lazy update algorithm for ODAMV? Is it a straightforward application of previous techniques?

4. In Conjecture 3.1 on HMV, is the time complexity of Phase 1 and Phase 2 the time it takes to read the input matrices?

5. Correct me if I am wrong, as I am not a complexity expert --- Below Conjecture 3.1, I think “reduce OAMV and ODAMV to HMV” should be “reduce HMV to OAMV and ODAMV”. Because the goal here is to show OAMV and ODAMV are harder to solve than HMV (in terms of computational complexity), and so the LB on computational complexity holds for HMV also holds for OAMV / ODAMV.
Nevertheless, the contradiction-based LB proof shown in Section 3.2 and 4 makes sense to me.

6. It is common to compute an approximation to the attention matrix (for faster run time). Do the techniques (algorithm and the lower bound) developed in this work extensible to dynamically maintain an approximation to the attention matrix?

---

> ### Author Response · Authors · 2023-11-21
> **Rebuttal by Authors - Part 1**
>
> **Q1**: During the training or deployment of an LLM, for example, the input sequences (of length $n$ ) are different per sample. Hence, the matrices $Q, K, V$ are different for each sample, and it is hardly the case that the attention matrix will be updated by only one entry at a time. Is it possible to give a practical scenario where the problem of dynamic/online attention maintenance with entry-wise updates considered in this work applies?It might be more interesting to consider a row or a column, or even a submatrix of fixed size, being updated in each iteration
>
> **A1**: Your careful reading and the insightful suggestions are greatly appreciated. In response to the concerns raised regarding the practical implications of the dynamic attention maintenance problem, we emphasise that entry-wise update of the weight matrix is prevalent in real-world scenarios. Deep neural network architectures frequently exhibit significant redundancy, and empirical evidence supports the capacity of deep neural networks to tolerate substantial levels of sparsity [1,2]. In downstream fine-tuning tasks, the dimensions of the model often make the fine tuning infeasible.
>
> Sparse neural networks have demonstrated enhanced parameter and computational efficiency relative to dense networks, and, in many cases, can significantly decrease wall clock inference times [3,4]. Over the past few years, numerous techniques for inducing sparsity have been proposed to sparsify the neural network such as magnitude pruning [1,5], RegL [6] and dynamic sparse reparameterization [7]. For attention based models, leveraging the sparsity of the attention can help make the model more efficient, as in [8,9,10].
> For example, to save the computation time and memory of the transformer, [11] only computes the top-k entries for each row in the attention matrix. Their work implies that we do not need to update the entire query and key matrix. A promising avenue for future exploration involves the adaptation of this concept to implement rank-1 updates, potentially surpassing entry-wise updates in effectiveness. We believe that it is an interesting direction for enhancing practical training efficiency. Our approach acknowledges the value of simplicity as an initial foundation, drawing from a range of theoretical papers [12,13,14,15]. Starting with a simpler version allows us to build a solid starting point for understanding the broader context.
>
> [1] Han, Song, et al. "Learning both weights and connections for efficient neural network." Advances in Neural Information Processing Systems 28 (2015).
>
> [2] Gale, Trevor, Erich Elsen, and Sara Hooker. "The state of sparsity in deep neural networks." arXiv preprint arXiv:1902.09574 (2019).
>
> [3] Theis, Lucas, et al. "Faster gaze prediction with dense networks and Fisher pruning." arXiv preprint arXiv:1801.05787 (2018).
>
> [4] Kalchbrenner, Nal, et al. "Efficient neural audio synthesis." International Conference on Machine Learning. PMLR, 2018.
>
> [5] Zhu, Michael, and Suyog Gupta. "To prune, or not to prune: exploring the efficacy of pruning for model compression." arXiv preprint arXiv:1710.01878 (2017).
>
> [6] Evci, Utku, et al. "Rigging the lottery: Making all tickets winners." International Conference on Machine Learning. PMLR, 2020.
>
> [7] Mostafa, Hesham, and Xin Wang. "Parameter efficient training of deep convolutional neural networks by dynamic sparse reparameterization." International Conference on Machine Learning. PMLR, 2019.
>
> [8] Roy, Aurko, et al. "Efficient content-based sparse attention with routing transformers." Transactions of the Association for Computational Linguistics 9 (2021): 53-68.
>
> [9] Choromanski, Krzysztof, et al. "Rethinking attention with performers." arXiv preprint arXiv:2009.14794 (2020).
>
> [10] Kaiser, Łukasz, and Samy Bengio. "Discrete autoencoders for sequence models." arXiv preprint arXiv:1801.09797 (2018).
>
> [11] Gupta, Ankit, et al. "Memory-efficient Transformers via Top-$ k $ Attention." arXiv preprint arXiv:2106.06899 (2021).
>
> [12] Alman, Josh, and Zhao Song. "Fast attention requires bounded entries." arXiv preprint arXiv:2302.13214 (2023).
>
> [13] Deng, Yichuan, Sridhar Mahadevan, and Zhao Song. "Randomized and deterministic attention sparsification algorithms for over-parameterized feature dimension." arXiv preprint arXiv:2304.04397 (2023).
>
> [14] Kitaev, Nikita, Łukasz Kaiser, and Anselm Levskaya. "Reformer: The efficient transformer." arXiv preprint arXiv:2001.04451 (2020).
>
> [15] Katharopoulos, Angelos, et al. "Transformers are rnns: Fast autoregressive transformers with linear attention." International conference on machine learning. PMLR, 2020.

---

> ### Author Response · Authors · 2023-11-21
> **Rebuttal by Authors - Part 2**
>
> **Q2**: In the abstract, $\omega(1,1, \tau)$ and $\tau$ are undefined notations that make it hard for the readers to understand the core results of the paper at first glance. While one might be able to infer that $\omega$ means the matrix multiplication exponent from the context, it is hard to understand $\omega(1,1, \tau)$. Also, without properly introducing $\tau$, it is unclear whether the run time results hold for a specifically chosen $\tau$ (might be too complicated to write out in the abstract) or for any $\tau \in(0,1]$.
>
> **A2**: Thanks for your careful reading and pointing them out. To fix the presentation issues, we have defined $\omega(1,1, \tau)$ to be a matrix multiplication exponent and $\tau$ to be a constant in $(0,1]$ in the abstract.
>
> **Q3**: In Theorem 1.3, $\delta$ is undefined. I was thinking about some notion of failure probability when I first read this, but it turns out in later sections that $\delta$ is indeed the update to the attention matrix.
>
> **A3**: Thanks for your careful reading and pointing them out. We have defined $\delta$ to be the update of the matrix in Theorem 1.3.
>
> **Q4**: Section 3.1 does not give a precise and clear overview of the lazy update based algorithm that solves ODAMV. For example, it is unclear what "List_C, List_D, List_V" in "Lazy Update" store. In "Fast Query", it is confusing as why there are two stacked matrices, $\Delta_{V, 1}$ and $\Delta_{V, 2}$ from List_V.
>
> **A4**: Thanks for your careful reading. We kindly argue that, as you can see in Line 16,17 of Algorithm 2 and Line 5 of Algorithm 3, the update that store in  "List_C, List_D, List_V" is a bit lengthy to write them down in technique overview. In Section 3.1, we just want to give reader a brief idea of how our algorithm works.
>
>
> **Q5**: In Theorem 1.3, "This operation has the same ... as K update" should be "This operation has the same ... as UpdateK"? Below Definition 1.2, "When then complement our result ..." => "We then complement our result"
>
> **A5**: Thanks for pointing this out. We have fixed them as you suggested.
>
> **Q6**: In Section 3.2 and 4 on the Lower Bound (LB), it is unclear why this work presents the LB result for a simpler problem called OAMV, which is not the true problem ODAMV considered, instead of directly presenting the LB results of ODAMV. Section 4 is essentially a more detailed / formalized version of Section 3.2. These two sections might be combined.
>
> **A6**: Due to space limitations, we provide our result and proof sketch for the simplified version of ODAMV, which is OAMV, in the main content and defer the proof of the true problem ODAMV to appendix. Section 3.2 is the technique overview that gives a proof sketch of our main results and Section 4 is the formal results for the OAMV problem.
>
> **Q7**:  On the algorithm / upper bound side, is there any utility guarantee of the algorithm? For example, given a query to the $(i, j)$-th entry, how close it is between $\widetilde{B}$ and $B$, where $B$ is the target matrix after updates (as defined in Section 3.1) and $\widetilde{B}$ is the updated matrix computed by the algorithm?
>
> **A7**: We do not use any approximation techniques like sampling, so the output of the data structure is identical to the true target matrix.
>
> **Q8**: A follow-up question: why does one need to recompute the attention matrix every $n^\alpha$ updates? Is it because after that many updates, certain utility guarantee no longer holds?
>
> **A8**: Yes, you are correct.
>
> **Q9**: What is the algorithmic challenge in designing the lazy update algorithm for ODAMV? Is it a straightforward application of previous techniques?
>
> **A9**: Thank you for your insightful question. It is essential to emphasize that our approach to ODAMV, while leveraging known techniques, is not merely a straightforward application of these methods. The intricacies lie in the specific adaptation and application to the ODAMV problem. The main purpose of the area of “fine-grained complexity theory” and their conditional lower bounds is to figure out the nature of the hardness, i.e. which problems boil down to the same source of difficulty/fundamental bottlenecks. E.g. problems with hardness from the OV (orthogonal vector) conjecture boil down to the fundamental bottleneck of searching, hardness from the BMM (boolean matrix multiplication) conjecture show that hardness comes from matrix multiplication, and problems with hardness from the HMV conjecture boil down to the trade-off between matrix-vector multiplication vs fast matrix multiplication. We show that dynamic attention maintenance belongs to the latter class by providing tight bounds.
>
> **Q10**: In Conjecture 3.1 on HMV, is the time complexity of Phase 1 and Phase 2 the time it takes to read the input matrices?
>
> **A10**: The time complexity of phase 1 and 2 is the time spent by an algorithm on processing the input. That includes reading the input matrices, but also any additional processing done during this phase.

---

> ### Author Response · Authors · 2023-11-21
> **Rebuttal by Authors - Part 3**
>
> **Q11**:  Below Conjecture 3.1, I think "reduce OAMV and ODAMV to HMV" should be "reduce HMV to OAMV and ODAMV". Because the goal here is to show OAMV and ODAMV are harder to solve than HMV (in terms of computational complexity), and so the LB on computational complexity holds for HMV also holds for OAMV / ODAMV. Nevertheless, the contradiction-based LB proof shown in Section 3.2 and 4 makes sense to me.
>
> **A11**: Thank you for pointing this out. It should have been “reduce HMV to AMV and ODAMV” and we have updated that.
>
> **Q12**: It is common to compute an approximation to the attention matrix (for faster run time). Do the techniques (algorithm and the lower bound) developed in this work extensible to dynamically maintain an approximation to the attention matrix?
>
> **A12**: Yes, and we believe that the adaptation of this concept to implement rank-1 updates will potentially surpasses entry-wise updates in effectiveness.

---

### Meta-Review · Area_Chair_g2XS · 2023-12-14

**Metareview:**

The paper studies a specific algorithmic problem arising in LLMs (mostly transformers) i.e. attention matrix multiplication. The object that needs to be computed is A = exp(QK^T) and the AV^T is output. The paper considers a dynamic setting where Q is fixed, but K and V can be updated and the goal is to have efficient updates and queries. Overall there was interest in this paper, but it was also felt that the contributions were mostly obtained by putting together several results (recent and old) from work in TCS.

**Justification For Why Not Higher Score:**

Relevance less clear, seems a more theory paper -- but the theory is about dynamic algorithms than ML. The relevance to ML is only through the origin of problem.

**Justification For Why Not Lower Score:**

NA

---

### Decision · Program_Chairs · 2024-01-16

Reject